_Article_

# Real-time assessment of mitochondrial DNA heteroplasmy dynamics at the single-cell level

Rodaria Roussou[1,2], Dirk Metzler [ID][1], Francesco Padovani [ID][3], Felix Thoma [ID][1,2], Rebecca Schwarz[1], Boris Shraiman[4], Kurt M Schmoller [ID][3] & Christof Osman [ID][1✉]

## Abstract

**Mitochondrial DNA (mtDNA) is present in multiple copies within cells and is required for mitochondrial ATP generation. Even within individual cells, mtDNA copies can differ in their sequence, a state known as heteroplasmy. The principles underlying dynamic changes in the degree of heteroplasmy remain incompletely understood, due to the inability to monitor this phenomenon in real time. Here, we employ mtDNA-based fluorescent markers, microfluidics, and automated cell tracking, to follow mtDNA variants in live heteroplasmic yeast populations at the single-cell level. This approach, in combination with direct mtDNA tracking and data-driven mathematical modeling reveals asymmetric partitioning of mtDNA copies during cell division, as well as limited mitochondrial fusion and fission frequencies, as critical driving forces for mtDNA variant segregation. Given that our approach also facilitates assessment of segregation between intact and mutant mtDNA, we anticipate that it will be instrumental in elucidating the mechanisms underlying the purifying selection of mtDNA.**

**Keywords** mtDNA; Heteroplasmy; Mitochondrial Fission; Mitochondria; Mathematical Modeling
**Subject Categories** Computational Biology; Organelles

## Introduction

The mitochondrial genome (mtDNA) encodes core subunits of the respiratory chain, making it essential for mitochondrial ATP production. Most eukaryotic cells contain multiple copies of mtDNA. From yeast to human cells, these copies are distributed throughout the mitochondrial network in a semiregular manner, with each mtDNA copy located roughly equidistantly from adjacent copies (Iborra et al, 2004; Jajoo et al, 2016; Osman et al, 2015). mtDNA is packaged into nucleoprotein complexes known as nucleoids (Chen and Butow, 2005; Spelbrink, 2010), containing on average 1–3 copies of mtDNA in mammalian and yeast cells

(Kukat et al, 2011; Seel et al, 2023). Mutations in individual mtDNA copies can result in heteroplasmy, where different mtDNA variants coexist within a cell (Stewart and Chinnery, 2021). This scenario can occur with neutral mutations having no impact on mitochondrial function, but becomes relevant for cellular and organismal health when mutations affect mitochondrial function (Park and Larsson, 2011). In this latter case, the ratio of mutant over intact mtDNA amounts in combination with the nature of the mutation determines the severity of the mutant phenotype.

Throughout the lifespan of an organism, the degree of heteroplasmy, representing the proportion of distinct mtDNA variants within individual cells, can change (Kauppila et al, 2017). The proposed mechanisms underlying this change include relaxed replication and vegetative segregation. Relaxed replication, wherein the amplification of mtDNA is uncoupled from the cell cycle, may lead to some mtDNA copies undergoing multiple replications while others remain unreplicated. Uneven distribution of mtDNA copies between mother and daughter cells, known as vegetative segregation, can further alter the degree of heteroplasmy in proliferating cell populations (Birky, 1994; Stewart and Chinnery, 2021). Additionally, intracellular selection resulting from favoring or disfavoring specific mtDNA variants for replication or degradation (Floros et al, 2018; Hill et al, 2014; Jakubke et al, 2021; Lieber et al, 2019; Ma et al, 2014) or selection acting on the level of cellular fitness (Kotrys et al, 2024) have been proposed to contribute to alterations in heteroplasmic levels in cell populations. Striking examples of changes in heteroplasmy are observed during germline development, where ratios of different mtDNA variants can rapidly shift from one generation to the next (Ashley et al, 1989; Burgstaller et al, 2018; Lee et al, 2012; Olivo et al, 1983; Otten et al, 2016; Wei et al, 2019). This rapid change is proposed to occur due to a bottleneck effect in oocyte development, either through partitioning of a limited number of mtDNA into primordial germ cells or preferential replication of a subset of mtDNA copies in developing oocytes (Cao et al, 2009; Cree et al, 2008; Otten et al, 2016; Stewart and Larsson, 2014; Wai et al, 2008).

The concept of heteroplasmy is widespread among eukaryotic cells. In the single-celled budding yeast, the degree of heteroplasmy in individual cells changes during the growth of a population derived from a single heteroplasmic zygote (Dujon et al, 1974). Eventually, this results in the complete segregation of mtDNA

[1]Faculty of Biology, Ludwig-Maximilians-Universität München, 82152 Planegg-Martinsried, Germany. [2]Graduate School Life Science Munich, 82152 Planegg-Martinsried, Germany. [3]Institute of Functional Epigenetics, Molecular Targets and Therapeutics Center, Helmholtz Zentrum München, 85764 Neuherberg, Germany. [4]Kavli Institute for Theoretical Physics, University of California, 93106 Santa Barbara, CA, USA. ✉E-mail: osman@bio.lmu.de

variants (Birky, 1994, 2001; Jakubke et al, 2021), such that most cells retain only one variant present in the heteroplasmic founder cell. So far, studying heteroplasmic dynamics has remained challenging regardless of the organism, because it has not been possible to infer the quantity of mtDNA variants within living cells over time. However, theoretical modeling has supported the view that heteroplasmy variance increases with cell divisions (Johnston, 2019; Johnston et al, 2015). Furthermore, modeling has also suggested that mitochondrial network structure and fission-fusion dynamics contribute to the emergence of heteroplasmic variance (Glastad and Johnston, 2023; Kowald and Kirkwood, 2011; Tam et al, 2013).

Here, we present an important step forward in understanding the temporal dynamics of mtDNA heteroplasmy in proliferating cell populations at single-cell resolution in *S. cerevisiae*. By employing two mtDNA variants encoding Atp6 fused to either red or green fluorescent proteins as surrogate markers for neutral mtDNA haplotypes in *S. cerevisiae*, we infer quantities of mtDNA variants in living cells. Using this approach, we observe a rapid shift in heteroplasmy levels across populations and find that asymmetric partitioning of mtDNA copies during cell division, as well as mitochondrial fusion-fission frequencies, are important determinants for mtDNA variant segregation.

# Results

## Real-time single-cell tracking of mtDNA heteroplasmy dynamics in *S. cerevisiae*

To investigate the dynamics of heteroplasmy, we aimed to develop a system, which would allow real-time assessment of the segregation of two different mtDNA genotypes across multiple generations. For this purpose, we opted for two recently established yeast strains with opposing mating types, each homoplasmic for a different mtDNA variant. Specifically, these strains harbor mtDNA encoding Atp6 tagged C-terminally either with NeonGreen (mtDNA$^{Atp6-NG}$) or mKate2 (mtDNA$^{Atp6-mKate2}$) and hence exhibit green or red fluorescence within mitochondria, respectively, depending on the presence and expression of the respective mtDNA haplotype. According to our rationale, the mating of both strains and subsequent mitochondrial fusion would result in heteroplasmic zygotes, carrying ~50% mtDNA$^{Atp6-NG}$ and ~50% mtDNA$^{Atp6-mKate2}$ (Jakubke et al, 2021; Nunnari et al, 1997; Okamoto et al, 1998). Upon growth, each heteroplasmic zygote would give rise to progeny, whose heteroplasmic state would be reflected by ongoing expression and fluorescence of Atp6-NG and/or Atp6-mKate2 (Fig. 1A). In the following analyses, the heteroplasmy state of each cell will be characterized by an "h-value," calculated by dividing the amount of mtDNA$^{Atp6-mKate2}$ by the sum of mtDNA$^{Atp6-NG}$ and mtDNA$^{Atp6-mKate2}$, inferred by the respective fluorescent signals (Fig. 1B). Unless otherwise stated, for all experiments, cells were pre-grown in glucose-containing rich medium, because of higher mating efficiency, while imaging was performed in minimal medium containing glucose, due to lower background fluorescence.

In order to continuously monitor heteroplasmy changes on a single-cell level, we used a microfluidic system, that allowed tracking of progeny derived from single zygotes for up to six generations. Segmentation and tracking of individual cells as well as lineage tracing enabled the construction of genealogical trees for every single population (Fig. 1C,D).

## Characterization of haploid mtDNA$^{Atp6-NG}$ and mtDNA$^{Atp6-mKate2}$ parental strains

Before generating heteroplasmic zygotes, it was important to compare growth and mtDNA maintenance between strains harboring mtDNA$^{Atp6-NG}$ or mtDNA$^{Atp6-mKate2}$ to determine potential influences of growth properties on segregational dynamics of both mtDNA variants in the following heteroplasmic segregation experiments. In line with previous analyses (Jakubke et al, 2021), both strains exhibited virtually wildtype-like growth under fermentable and non-fermentable carbon sources at 30 and 37 °C in plate growth assays (Appendix Fig. S1A,B). We furthermore quantified doubling times of mtDNA$^{Atp6-NG}$ or mtDNA$^{Atp6-mKate2}$ strains in our microfluidic setup by segmenting and counting cells over a time course of 8 h. In line with our growth assay on plates, both strains exhibited very similar doubling times (Appendix Fig. S1C). Similarly, petite levels, indicative of loss or dysfunctional mtDNA, were equivalent between mtDNA$^{Atp6-NG}$, mtDNA$^{Atp6-mKate2}$, and wildtype strains (Appendix Fig. S1D).

Next, we aimed to assess the validity of using Atp6-NG or Atp6-mKate2 fluorescence intensities as proxies for mtDNA presence within single cells. First, we examined mtDNA$^{Atp6-NG}$ or mtDNA$^{Atp6-mKate2}$ strains in our microfluidic setup and monitored fluorescent signals across multiple generations for 8 h. In every population derived from single haploid cells, we observed homogeneous fluorescent signal intensities in the mitochondrial networks across all cells after 8 h (Fig. 2A,B; Movies EV1, EV2). Next, we assessed Atp6-NG and Atp6-mKate2 fluorescence in cell populations lacking the mitochondrial HMG-box protein Abf2. Δ*abf2* cells are known to gradually lose mtDNA if grown in a fermentable carbon source (YPD) (Schrott and Osman, 2023; Sia et al, 2009; Zelenaya-Troitskaya et al, 1998). We cultured Δ*abf2* cells in a non-fermentable carbon source (YPG) to prevent mtDNA loss and monitored Atp6-NG and Atp6-mKate2 fluorescence upon transition to a minimal medium. Consistent with the occasional loss of mtDNA, Δ*abf2* populations exhibited indeed a higher signal variability across cells, which is quantitatively revealed by a higher coefficient of variation for either mtDNA$^{Atp6-NG}$ or mtDNA$^{Atp6-mKate2}$ in Δ*abf2*, compared to the respective WT cells (Fig. 2C,D; Appendix Fig. S2; Movies EV3, EV4). Some cells exhibited fluorescence, indicative of mtDNA presence, while others appeared to have fully lost mtDNA, as they were only visible in the DIC channel. We interpret these findings to reflect that all wildtype cells maintain mtDNA and continue to express the fluorescent Atp6 variants with relatively little cell-to-cell variability, while mtDNA maintenance deficits associated with Δ*abf2* cells result in heterogeneity of Atp6-NG or Atp6-mKate2 fluorescence signals.

To further support that Atp6-NG or Atp6-mKate2 fluorescence reflects the expression of the respective mtDNA, we measured the fluorescent decay rates by inhibiting mitochondrial translation with Chloramphenicol (CAP). Of note, this experiment was performed in strains lacking *PDR5*, a multidrug transporter localized in the plasma membrane, to prevent CAP export from the cell and allow efficient translational inhibition (Leonard et al, 1994). *PDR5*

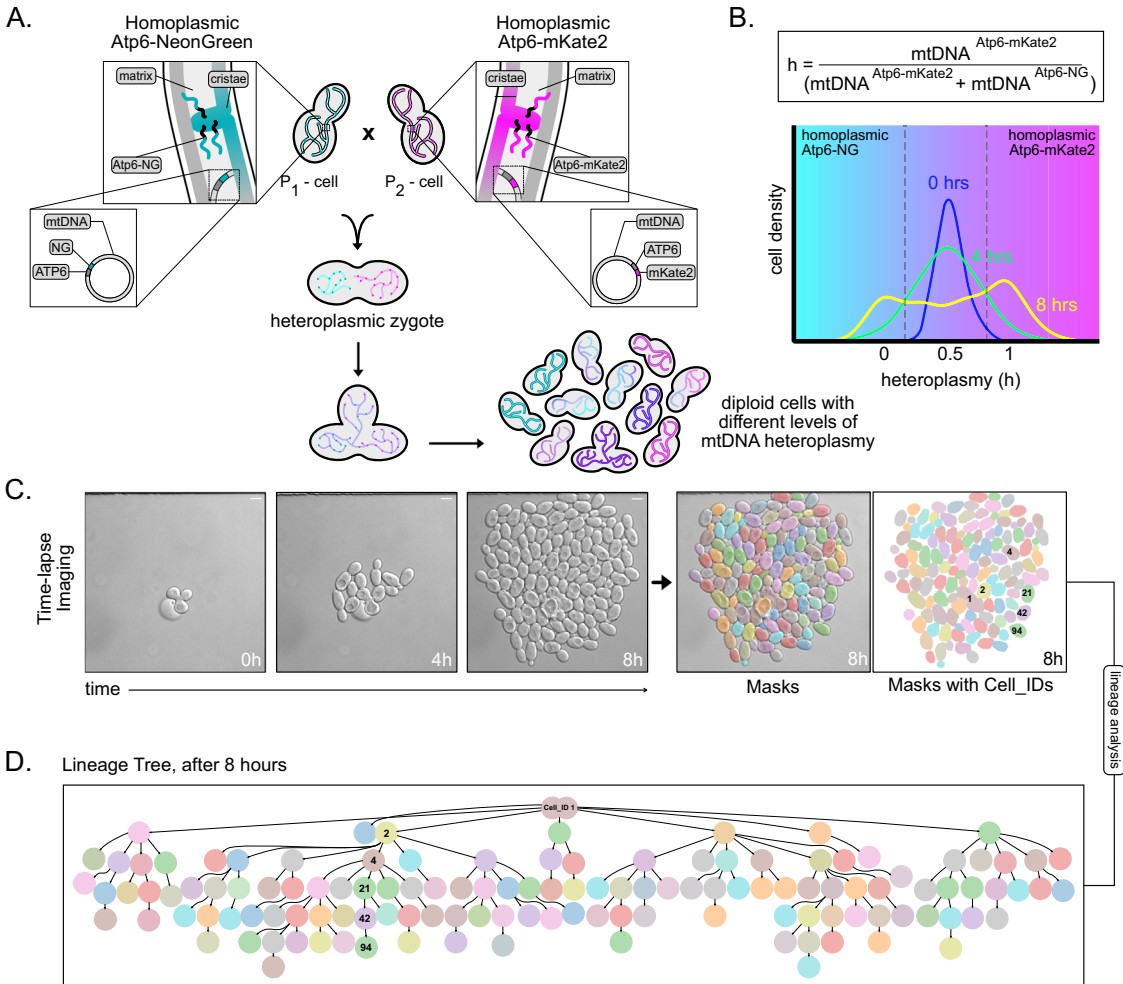

**Figure 1. Schematic of the mtDNA heteroplasmy pipeline.**

(A) Strains harboring either mtDNA$^{Atp6\text{-NG}}$ or mtDNA$^{Atp6\text{-mKate2}}$ are mated. The growth of a population derived from single zygotes is monitored in a microfluidic chamber, resulting in diploid cells with different levels of mtDNA heteroplasmy. (B) The heteroplasmy value (h) of each cell is calculated by dividing the fluorescent signal of Atp6-mKate2 by the sum of Atp6-NG and Atp6-mKate2. Example of heteroplasmy distributions at three indicated timepoints. Cells with h-values lower than 0.5 (cyan area) contain a higher proportion of mtDNA$^{Atp6\text{-NG}}$, while cells with h-values above 0.5 (magenta area) contain a higher proportion of mtDNA$^{Atp6\text{-mKate2}}$. (C) Representative bright-field images of a cell population derived from a single zygote, after an 8-h time-lapse video. Cell segmentation and masking was performed with the Cell-ACDC software. Scale bars: 5 µm. (D) Lineage tree derived from the zygote shown in (C), at the end of the recording. One lineage is annotated as an example, with cell identities matching the mask IDs in (C).

deletion did not lead to a growth phenotype (Appendix Fig. S1A,B). Upon addition of CAP, we observed a decrease in both Atp6-mKate2 and Atp6-NG intensities, where within 5 h, signal intensities appeared to reach a bottom plateau, likely representing background fluorescence of cells and absence of the respective fluorophore (Fig. 2E; Appendix Fig. S3A–D; Movies EV5–EV8). By fitting exponential decay curves (Alber et al, 2018), we assessed the fluorescence decay rates (k) for both fusion proteins, which were virtually identical ($k_{mKate2} = 0.31$, SEM:±0.01), $k_{NG} = 0.31$, SEM. ± 0.01) (Fig. 2F). Of note, in the absence of CAP, the wildtype mtDNA$^{Atp6\text{-NG}}$ strain exhibited an increase of fluorescence over time, while the mtDNA$^{Atp6\text{-mKate2}}$ strain exhibited a slight decrease over time. We also assessed the reappearance of fluorescent signals upon removal of CAP after a 6-h treatment. While both fluorescent signals reappeared rapidly, reflecting the maintenance of mtDNA, the Atp6-NG signal reemerged slightly faster compared to Atp6-

mKate2 (Figs. 2F and EV1; Movie EV9). We speculate the difference in fluorescence levels over time between Atp6-NG and Atp6-mKate2 to be the result of a combination of various factors, including the switch from rich to minimal medium, which may affect mtDNA expression, the biophysical properties of the fluorescent proteins, as well as differences in the maturation time between NG and mKate2. To account for the difference, in all following experiments, NG and mKate2 fluorescence is always normalized to the median fluorescence per timeframe in the respective channel.

Based on the aforementioned experiments, we conclude that the presence of mtDNA$^{Atp6\text{-NG}}$ or mtDNA$^{Atp6\text{-mKate2}}$ is unlikely to substantially alter cellular fitness, mtDNA maintenance, or influence the outcome of segregation dynamics within our heteroplasmic model. Furthermore, our findings support that

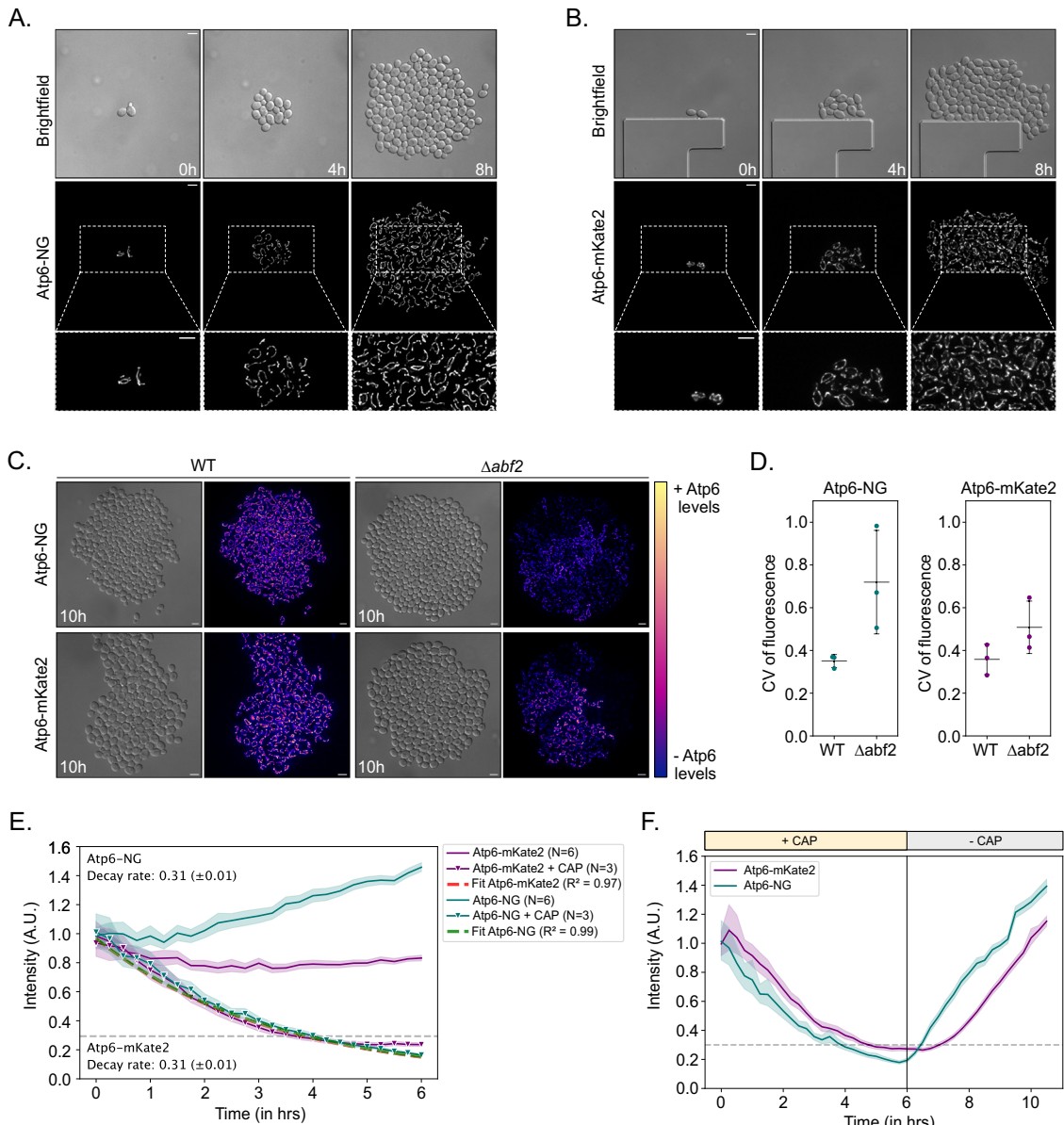

**Figure 2.  Atp6-NG and Atp6-mKate2 are valid proxies for mtDNA variants.**

(**A, B**) Yeast cells harboring Atp6-NG (**A**) or Atp6-mKate2 (**B**) mtDNA were imaged for 8 h on a microfluidic chip. Fluorescence images are maximum-intensity projections of z-stacks, after deconvolution. Scale bars: 5 μm. (**C**) Wildtype and Δ*abf2* cells harboring Atp6-NG or Atp6-mKate2 mtDNA were imaged for 10 h. Cells were first grown in YPG and supplied with minimal media during the microfluidic imaging. The color representation on the heatmap bar corresponds to the pixel intensity of Atp6. Fluorescence images are maximum-intensity projections of z-stacks, after deconvolution. Scale bars: 5 μm. (**D**) The coefficient of variation (CV) of fluorescence among cells at the last timepoint was calculated for indicated strains. Strains were pre-cultured in YPG and put into minimal media for microfluidic imaging. Each dot represents the CV of a population derived from single cells (*n* > 1000 cells analyzed in total per strain). The mean and SD of the CV per strain is shown (*N* = 3). (**E**) Line graph showing the normalized fluorescent intensities of Atp6-NG or Atp6-mKate2 assessed during growth of mtDNA$^{Atp6-NG}$ or mtDNA$^{Atp6-mKate2}$ strains. Cells were constantly supplied with minimal medium containing (*N* = 6) or lacking (*N* = 3) 1 mg/ml Chloramphenicol (CAP) for mtDNA translation inhibition. Intensities were normalized to the median fluorescence intensity of the first timepoint per replicate. Data represent mean and shaded areas show the 95% confidence interval per strain. Decay rates for each fluorophore were calculated by fitting an exponential to the intensity levels of fluorescence upon CAP treatment. (**F**) Line graph represents the normalized fluorescent intensities of cells expressing Atp6-NG or Atp6-mKate2, during a 12-hr recording. Cells were first incubated for 6 h in minimal media with 1 mg/ml CAP and upon washing off the drug, cells were kept in minimal media until the end of each experiment. Data represent the mean and the shadow areas show the 95% confidence interval per strain (*N* = 3). Source data are available online for this figure.

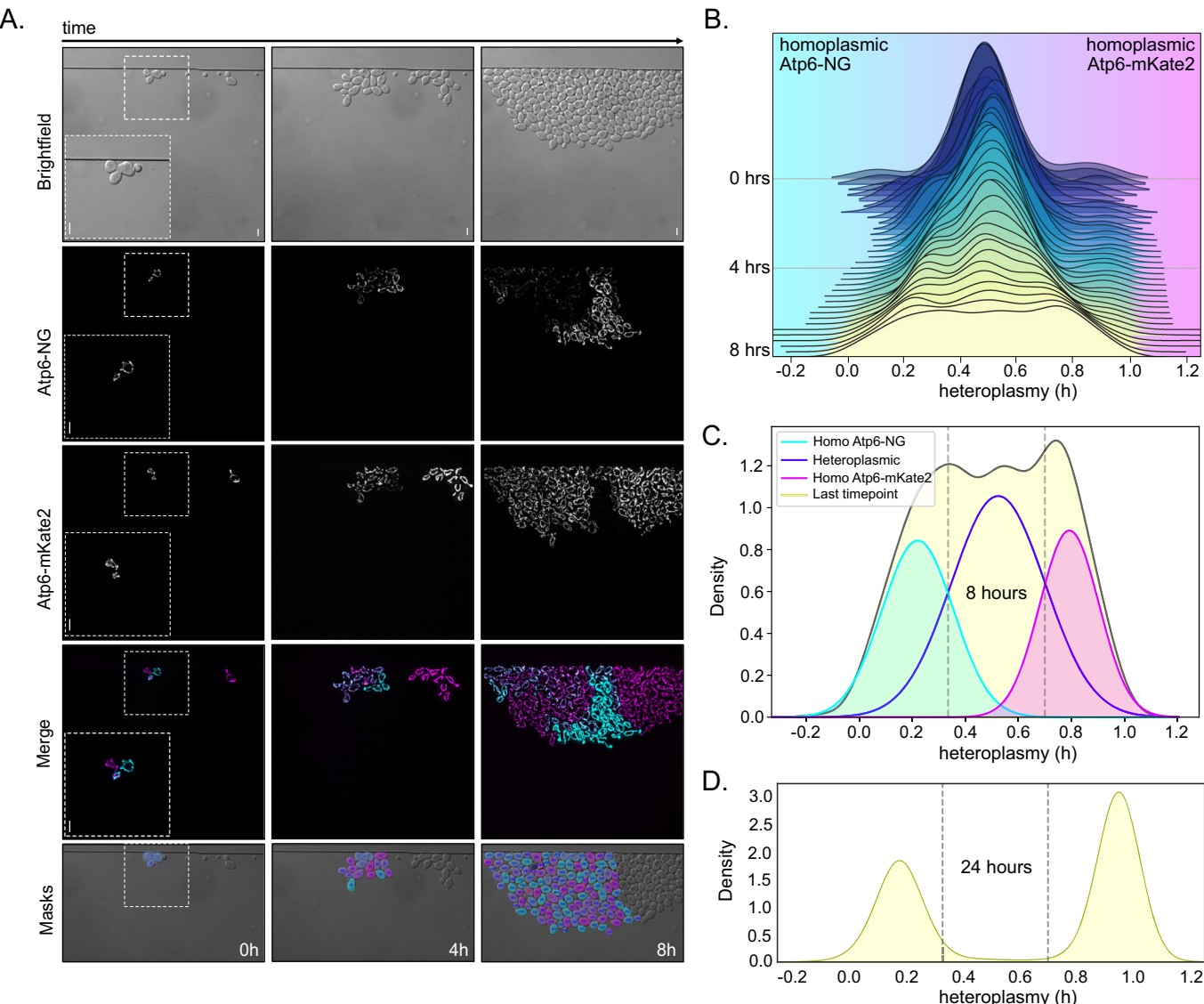

**Figure 3. Rapid shift of mtDNA content in heteroplasmic cell populations.**

(**A**) Representative images of a heteroplasmic population at indicated timepoints. Log-phase yeast cells harboring Atp6-NG (cyan) and Atp6-mKate2 (magenta) mtDNA were mated for 1.5 h prior to each microfluidic experiment. Individual heteroplasmic zygotes were chosen in a single field-of-view (FOV) and the growing cell populations were imaged for 8 h on a microfluidic chip. Fluorescence images are maximum-intensity projections of z-stacks, after deconvolution. Images representing the masks are derived from Cell-ACDC. Cells derived from a haploid cell present in the FOV were not segmented. Scale bars: 5 μm. (**B**) Joyplot of heteroplasmy levels of cells from a total of nine populations derived from heteroplasmic zygotes across the 8 h time-lapse recordings. (**C**) Single Gaussian curve fittings were applied on the three emerging peaks of the final cell distribution of the data at 8 h (density plot in light yellow, as in **B**). Three colored curves (cyan, purple, magenta) represent the cell distributions with more mtDNA^Atp6-NG (cyan), more mtDNA^Atp6-mKate2 (magenta), or cells harboring both (purple). Homoplasmy thresholds were set based on the intersection points of Gaussian curves. An h-value of 0.33 was set as the homoplasmy threshold for cells mainly harboring mtDNA^Atp6-NG and 0.71 as the threshold for mtDNA^Atp6-mKate2 homoplasmy, shown as gray dashed lines. (**D**) Density plot showing the heteroplasmy distribution of cells after 24 h. Atp6-NG and Atp6-mKate2 cells were mated for 2 h. Twenty individual zygotes were micro-dissected, upon growth on YPD plates for 24 h and heteroplasmic states of all cell populations were assessed by microscopy (N = 20 colonies, n = 7538 cells). Homoplasmy threshold values from the 8 h intersection points are shown in gray dashed lines. Source data are available online for this figure.

Atp6-NG and Atp6-mKate2 serve as valid proxies for the presence of the mtDNA by which they are encoded.

## Rapid segregation of mtDNA variants

Next, we followed the segregation of mtDNA^Atp6-NG and mtDNA^Atp6-mKate2 in populations derived from heteroplasmic

zygotes, generated by mating parental cells harboring either mtDNA variant (Fig. 1A; Movie EV10). As expected, all newly formed zygotes displayed a fused mitochondrial network that exhibited signals in both fluorescent channels, indicative of carrying both mitochondrial genomes. We proceeded to generate time-lapse videos for a total duration of 8 h to track the mtDNA segregation across multiple generations. Remarkably, cell-to-cell

heterogeneity became rapidly evident in all populations, reflecting different heteroplasmy levels (Fig. 3A; Appendix Fig. S4A). Most interestingly, patches of cells emerged in all populations that had transitioned to exhibiting fluorescence only in a single channel during the time course, indicating that these cells had retained predominantly or exclusively one of both mtDNA variants.

To quantitatively assess the heteroplasmy levels within each cell and population, we applied the heteroplasmy formula $h = m1/(m1 + m2)$, (Aryaman et al, 2019; Johnston et al, 2015), where m1 and m2 are the fluorescent intensities of Atp6-NG and Atp6-mKate2 per cell that were normalized to cell volume as well as median fluorescence per timeframe. Furthermore, the background signal present in cell-free areas of images was subtracted from cellular fluorescence. As a result, an h-value of 0.5 is indicative of an equal ratio of both mtDNA variants within cells, equidistant to the homoplasmy extremes, which tend towards 1 or 0 when there is a shift towards mtDNA$^{\text{Atp6-NG}}$ or mtDNA$^{\text{Atp6-mKate2}}$, respectively. By plotting the quantities of cells with specific h-values over time, we observed a shift in heteroplasmic states (Fig. 3B). Initially, the heteroplasmy distribution was narrowly centered around the average value of 0.5, where most cells exhibited fluorescence for both Atp6 variants. As time progressed, the middle peak flattened and the overall heteroplasmy distribution broadened (Fig. 3B). Focusing on the final timepoint, two peaks at both ends of the spectrum became apparent, closer to h-values of either 1 or 0 (Fig. 3C; Appendix Fig. S4B), likely representing homoplasmic cells.

Of note, for all segregation experiments, zygotes were chosen that gave rise to medial buds, which inherit mitochondria from both parental cells and, therefore, end up being heteroplasmic (Strausberg and Perlman, 1978). The inheritance of a mixed content derived from both parental cells is apparent from the green and red fluorescence within the mitochondria of medial buds (Appendix Fig. S5A). Second-daughter cells from zygotes often bud off of lateral positions and predominantly or exclusively inherit mitochondrial content from the parental cell from which they originate. Importantly, the segregation pattern did not significantly change, when we excluded cells derived from second-generation daughter cells that had appeared on lateral zygote positions and, therefore, had started out with predominantly only one of both mtDNA variants (Appendix Fig. S5B).

To estimate the proportions of homo- and heteroplasmic cells, we fit three Gaussian curves to the observed trimodal heteroplasmy distribution at the final timepoint. These curves characterize cells primarily exhibiting homoplasmy for either mtDNA$^{\text{Atp6-NG}}$ (on the left) or mtDNA$^{\text{Atp6-mKate2}}$ (on the right), as well as heteroplasmic cells (in the middle). While definitively distinguishing between homo- and heteroplasmy is not feasible, we approximated their quantities by establishing thresholds at the intersections of individual Gaussian curves. Cells falling below an h-value of 0.35 or above 0.7 were classified as homoplasmic for mtDNA$^{\text{Atp6-NG}}$ or mtDNA$^{\text{Atp6-mKate2}}$, respectively (Fig. 3C). This approximation is further supported by the fact that residual non-mated haploid cells homoplasmic for either mtDNA$^{\text{Atp6-NG}}$ or mtDNA$^{\text{Atp6-mKate2}}$ exhibited h-values comparable to diploid cells classified as homoplasmic (Appendix Fig. S6). We posit that the homoplasmic peaks not being centered around 0 and 1 is primarily attributable to autofluorescence in the "absent" channel. To confirm that diploid homoplasmic cells indeed fall below or above these thresholds, we assessed heteroplasmic states of 20 independent populations

derived from heteroplasmic zygotes after 24 h, where we expected virtually full segregation of mtDNA variants. Spatial limitations within the microfluidic system prevented its use for this approach. Therefore, we opted for a microdissection of heteroplasmic zygotes and observed mtDNA segregation in single cells after 24 h of growth on plates (Fig. EV2). Indeed, the 24 h cell distribution appeared bimodal, where the middle curve, representing heteroplasmic cells, was virtually absent (Fig. 3D). The two pronounced peaks fell below or above our homoplasmy thresholds, establishing them as good estimates to quantify homoplasmic cells. Based on these cutoffs, 97% of cells were characterized as homoplasmic, revealing, as expected, nearly complete segregation of two mtDNA variants within a growing yeast population after 24 h. Thus, our approach allows real-time imaging of mtDNA segregation dynamics at a single-cell level. Our analysis reveals rapid segregation, which can be observed already during the first 8 h and virtually completes within 24 h.

## Mathematical modeling of mtDNA segregation dynamics

To better understand how the observed mtDNA segregation occurs, we developed a mathematical model that simulates the dynamics of segregation in a growing cell population arising from a single heteroplasmic cell. In vivo, mtDNA is distributed throughout a tubular mitochondrial network within cells and partitioning of mitochondria, including mtDNA, occurs via the transport of mitochondrial tubules into daughter cells during cell division (Osman et al, 2015). Hence, mtDNA copies present in the same segments of the tubular mitochondrial network are more likely to be segregated together into daughter or mother cells, respectively. Therefore, a simple random-pick model would not adequately capture the in vivo situation. To reflect this morphological aspect in the model, we used arrays, representing the tubular organization of mitochondria that contained two mtDNA variants either denoted as 0 or 1. Of note, we do not consider branching of the array. The length of this array represents the previously determined average of 32 mtDNA copies per diploid cell (Göke et al, 2020), neglecting the cell-to-cell variability in the copy numbers that may occur due to cell size differences. Importantly, we simulated zygotes with equal amounts of 0 and 1 s in an initial 'mixed' structure sequence of 01010101… (Fig. 4A). We substantiated the model's robustness through manipulation of the array, by mildly varying the mtDNA copy numbers between 26 and 38 (Appendix Fig. S7A), as well as by substituting the initial structure of the founder cell with a semi-mixed array ("0011100011…") or a non-mixed array ("0000011111…") (Appendix Fig. S7B,C).

To account for mitochondrial fission and fusion events, we introduced the *nspl* parameter, which represented the number of fragments an array could split into, followed by re-fusion in random order during growth. Upon completion of this shuffling process, the array was allowed to split into two parts, which partitioned to mother and daughter cells. The daughter cell received the shorter of both parts containing a specific amount of mtDNA copies, defined by the *ndau* parameter (Fig. 4A). Depending on the inherited mtDNA variants, each daughter cell consisted of either all 0 s, all 1 s, or a mixture of both. Upon division, mtDNA copies were allowed to replicate until the total number reached 32. To reflect relaxed mtDNA replication, we randomly chose mtDNA molecules for replication, allowing multiple rounds of replication

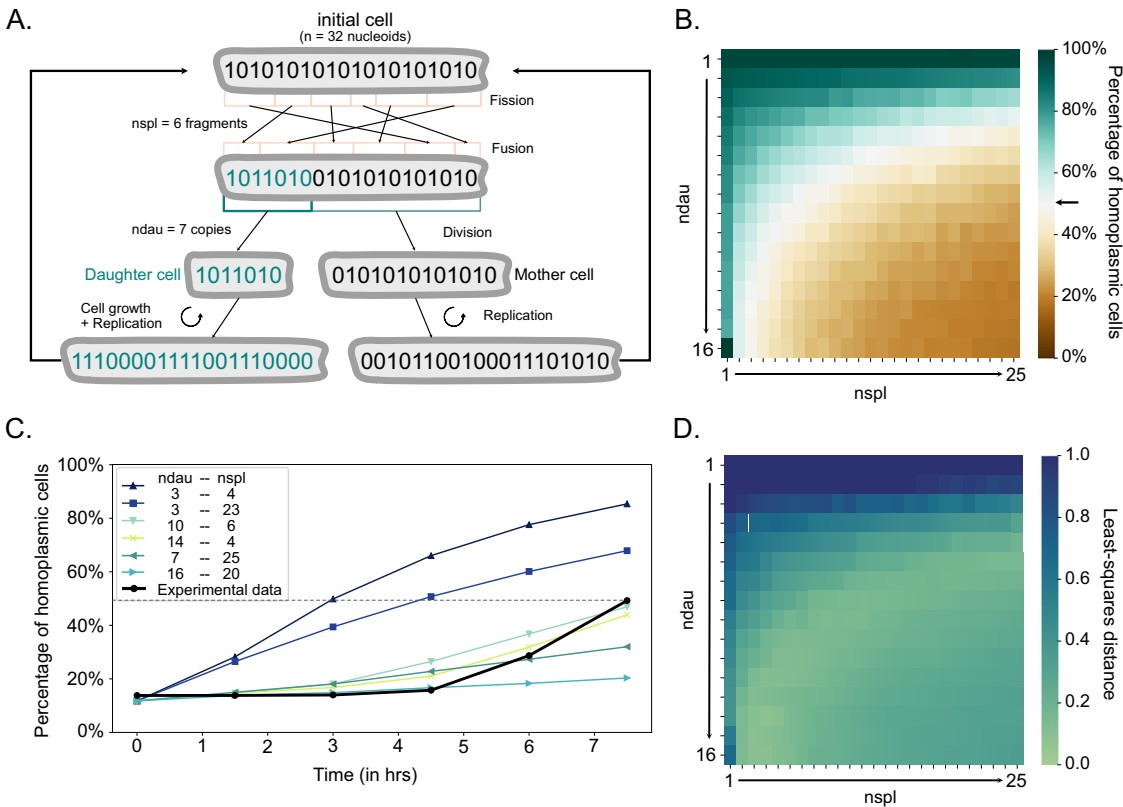

Figure 4.   Mathematical modeling of mtDNA segregation dynamics.

(A) Mitochondria are simulated as arrays, where 0 and 1s represent different mtDNA variants. Each single cell contains one such array with 32 mtDNA copies. The parameter *nspl* mimics the fission-fusion dynamics and the *ndau* parameter describes the number of mtDNA copies transferred to the daughter cell upon division. Each cell is allowed to give rise to a daughter cell once it reaches 32 mtDNA copies, upon random sequential replication of the inherited 0 and 1s. In the shown example, where *nspl* = 6, the array splits into six fragments and then stochastically fuses back together. For simplicity, only 20 copies are shown. (B) Heatmap showing the percentage of homoplasmic cells at timepoint *t* = 7.5 h, for different *ndau* and *nspl* pairs. The arrow indicates the proportion (50%) of cells being virtually homoplasmic in the empirical data, based on the pre-established homoplasmy cutoffs. Each simulation with any given combination of the two parameters has been run ten times. (C) Curves display the proportion of homoplasmic cells in experimental (black line) or simulated data derived from different *ndau* and *nspl* combinations (colored lines) for timepoint *t* = 0, 1.5, 3, 4.5, 6, 7.5 h. The dashed line represents 50% homoplasmy. (D) Heatmap showing the least-squares distance of simulations with a given pair of *ndau* and *nspl* parameters from the experimental data, upon application of the homoplasmy thresholds. Each simulation with any given combination of the two parameters has been run ten times. Source data are available online for this figure.

for the same mtDNA molecule during the span of one cell cycle. Notably, newly replicated mtDNA copies were always placed next to the template mtDNA. We neglected the possibility of spontaneous cell death and mtDNA degradation.

First, we simulated the influence of *nspl* and *ndau* parameters on the segregation rate by modeling the amount of homoplasmic cells after 7.5 h (Fig. 4B). We observed that higher *nspl* or *ndau* parameters, corresponding to more shuffling or more mtDNA copies partitioned to daughter cells, predict a relatively low percentage of homoplasmic cells after 7.5 h. Conversely, lower *nspl* and/or *ndau* values lead to an acceleration of mtDNA segregation. Our simulations show that distinct *ndau-nspl* combinations can result in similar levels of homoplasmic cells after 7.5 h. We next asked, which of these combinations aligns with our experimental data. To this end, we applied the previously defined homoplasmy thresholds (Fig. 3C) to all timepoints within our 8-hr in vivo experiment and quantified amounts of homoplasmic cells (Fig. 4C, black curve). We compared the fractions of homoplasmic cells between the experimental and the simulated data at timepoints 1.5,

3, 4.5, 6, and 7.5 h by applying the least-squares criterion. We found the best-fitting parameter values to be *ndau* = 14 and *nspl* = 4 (Fig. 4C,D). However, we identified various combinations of *ndau* and *nspl* approximating the experimental results. In particular, there was an inverse relationship between the values for *nspl* and *ndau* that fit the data well, where higher *ndau* values required lower *nspl* ones, and vice versa. Specifically, the combinations of relatively higher *ndau* values (ranging from 10-16) and lower *nspl* values (within the range of 3–6) came closest to the empirical data (Fig. 4C, light green lines). In contrast, we observed a poor fit to the data, when *ndau* and *nspl* were simultaneously very low or very high (Fig. 4D). Thus, these results suggest that fusion and fission frequencies, representing shuffling events, as well as the number of mtDNA copies transferred to daughter cells are contributing to the segregation kinetics of mtDNA variants in a yeast population, under our neutral experimental setup.

Next, we used our model to examine how increased mtDNA copy numbers within cells may affect mtDNA variant segregation. To this end, we simulated segregation in cells with 56 (observed in

cells with increased mtDNA copy number lacking the gene *MRX6* (Göke et al, 2020)) or 90 mtDNA copies (as an extreme example) (Fig. EV4A,B) and compared segregation kinetics to the kinetics from the best-fitting simulations for cells with 32 mtDNA copy numbers, where *ndau* and *nspl* equaled 14 and 4, respectively (Fig. EV4C–E). We found that the segregation speed is very similar if the *nspl* value stays the same (nspl=4) and the same percentage of total copies of mtDNA are passed on to the daughter cell (e.g., ~44% would mean 14 copies in a cell with 32 copies, but ~25 or ~39 copies in a cell with 56 or 90 copies, respectively) (Fig. EV4C). However, it is also possible that increased mtDNA copy number affects the *ndau* and/or *nspl* parameters. Similar to our simulations for mtDNA copy number equal to 32 (Fig. 4D), our simulations suggest that also in scenarios with increased mtDNA copy number, a higher percentage of mtDNA copies transmitted to daughter cells and increased fusion-fission frequencies could result in slower segregation of mtDNA variants. On the contrary, a lower percentage of transmitted copies and decreased fission-fusion frequencies would result in faster segregation (Fig. EV4D,E).

Altogether, these results suggest that fusion and fission frequencies, representing shuffling events, as well as the number of mtDNA copies transferred to daughter cells are contributing to the segregation kinetics of mtDNA variants in a yeast population, under our neutral experimental setup.

## Partitioning of a small subset of mtDNA copies to daughter cells promotes mtDNA homoplasmy

Our mathematical model suggests that multiple different combinations of *nspl* and *ndau* parameters could explain the observed segregation of mtDNA variants. With respect to the *ndau* parameter, the values range from seven to fifteen, representing about a fifth or half of available mtDNA copies passed on to the daughter cell, respectively. Therefore, we asked which of the possible model predictions reflects the in vivo situation. First, we examined our experimental heteroplasmy dataset to evaluate if equal or unequal amounts of mtDNA copies are partitioned to mother and daughter cells during cell division. If unequal amounts of mtDNA copies are transferred per division to the progeny, we would expect a lower correlation between heteroplasmy values between mothers (M) and daughter cells (D) compared to a correlation of h-values between mothers to themselves at a later timepoint ($M_D$). In a scenario where mtDNA copies are equally split between mother and daughter, the M-D and M-$M_D$ h-value correlations are expected to be similar. Hence, we compared heteroplasmy levels of mother cells (M) at timepoint $t_M$ to those of their daughters (D) at $t_D$ or granddaughters (GD) at $t_{GD}$, as well as to those of their aging selves $M_D$ and $M_{GD}$ at $t_D$ or $t_{GD}$, respectively (Fig. 5A). The two timepoints, $t_D$ and $t_{GD}$, were chosen based on the same growth stage, where cells had small buds (1/5 bud-to-mother volume ratio). In line with an unequal partitioning of mtDNA copies, we observed a high correlation between M and $M_D$ in contrast to a statistically significant lower correlation coefficient between M and D (Fig. 5B). Comparison of mothers to their granddaughters further supported this hypothesis, as the M-GD correlation further decreased compared to the M-D correlation. In contrast, aged mothers ($M_{GD}$) still displayed a stronger similarity to their original heteroplasmic state (M). These results underscore that cell division, most likely through the transmission of a limited

number of mtDNA copies, is a major driver for the progressive divergence of heteroplasmic states in a proliferating yeast population.

Subsequently, we aimed to directly determine the number of mtDNA copies passed from mother to daughter per cell division. Therefore, we employed the mtLacO-LacI system, which allows the detection of single mtDNA copies by fluorescence live-cell microscopy (Osman et al, 2015). First, we quantified the number of fluorescent foci, representing mtDNA copies, migrating across the bud neck from a mother to its bud in 5-minute windows. Cells were in different cell cycle stages, however only mother-bud pairs whose mitochondrial network appeared interconnected or where mitochondrial fragments traveled from mother to daughter were considered. We observed that, on average, 1.18 mtDNA foci were transmitted to daughter cells within these 5 min (Fig. 5C,D; see Methods; Appendix Fig. S8A,B; Movie EV11). Next, we examined the duration for which the exchange of mitochondrial content occurred between mother-daughter pairs, by live-cell microscopy using the nuclear-encoded matrix-targeted mKate2 (see Methods). We did not observe mitochondrial content exchange, neither through continuous mitochondrial tubules spanning the mother-bud neck, nor through transport of mitochondrial fragments, after an average time period of $\Delta t = 47.5$ min (Fig. 5E; Appendix Fig. S8C; Movie EV12). This observation aligns well with a previous study that similarly observed that daughter cells gain mitochondrial volume at the expense of the mother during the first half of the cell cycle (Rafelski et al, 2012). Taken together, we estimate that, on average, 11.2 nucleoids migrate from mother to daughter in a single-cell division (Fig. 5F).

To further examine mtDNA transmission to daughter cells, we performed live-cell microscopy of diploid cells containing the mtLacO-LacI system over the duration of an entire cell cycle and assessed dynamic changes of mtDNA content in virgin mother and emerging daughter cells. In line with our observation that the exchange of mitochondrial content occurs between mother and daughter cells for about 45 min, we observed a reduction of mtDNA foci from about 35 to below 30 in the mother cells during this time period, whereas the number of foci increased to 15 in daughter cells. After 45 min, the number of foci remained constant in the mother cell, while the foci number further increased to about 30 in the daughter cell (Fig. EV4). These results suggest that mtDNA copy number is replenished in mother cells, while mtDNA copies are being passed on to daughters, and that after 45 min mtDNA replication continues in daughter cells. Of note, we did not observe that the starting foci number of 35 was reestablished in the mother cell or the daughter cell. We assume that this observation is due to continued imaging causing phototoxicity or bleaching.

Besides asymmetric apportioning of mtDNA copies to daughter cells, our mathematical model also predicts a critical role for mitochondrial fusion and fission cycles in determining the rate of mtDNA variant segregation by shuffling mtDNA copies in the mitochondrial network. To experimentally test this prediction, we performed mtDNA variant segregation experiments in cells lacking the gene *DNM1*, which is essential for mitochondrial fission. Specifically, we mated Δ*dnm1* cells containing mtDNA^Atp6-NG or mtDNA^Atp6-mKate2, pre-grown in YPG medium to prevent mtDNA loss, and microscopically evaluated the percentage of heteroplasmic cells in colonies formed from individual zygotes after 18 h of growth on YPD plates. In line with our model, the

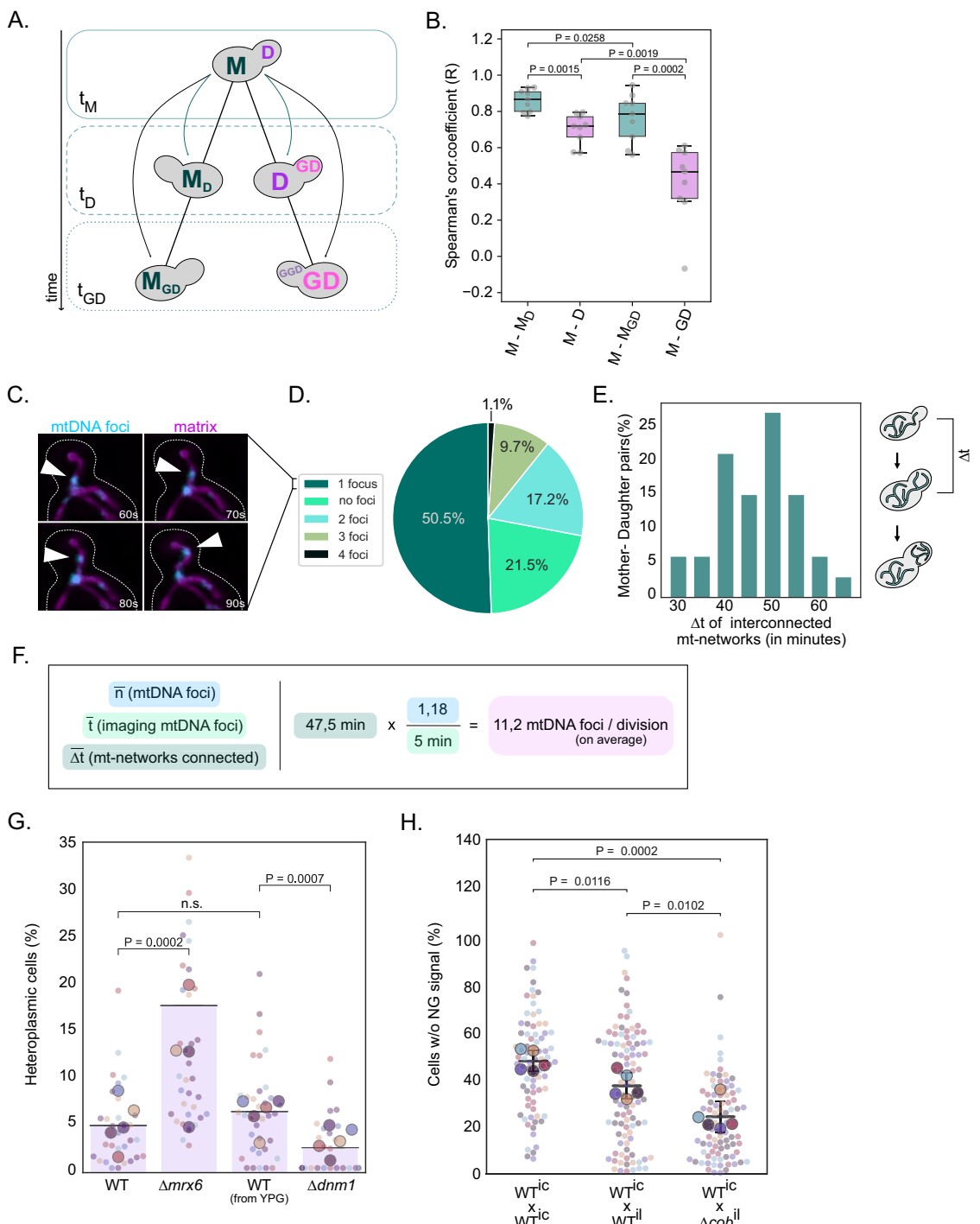

absence of mitochondrial fission resulted, on average, in a lower amount of heteroplasmic cells compared to mating between mtDNA$^{Atp6-NG}$ or mtDNA$^{Atp6-mKate2}$ WT cells, indicating faster segregation kinetics in the absence of mitochondrial fission (Fig. 5G). This conclusion is further supported by the observation that we detected exclusively homoplasmic cells in 57% of colonies derived from heteroplasmic Δdnm1 cells compared to 14% in WT cells (Appendix Fig. S9B).

We additionally experimentally tested the effect of increased mtDNA copy number on mtDNA segregation dynamics. To this end, we performed segregation experiments in Δmrx6 cells, that have previously been shown to have a twofold increase of mtDNA (Göke et al, 2020). We assessed the number of heteroplasmic cells in populations after growth for 18 h derived from Δmrx6 zygotes containing mtDNA$^{Atp6-NG}$ and mtDNA$^{Atp6-mKate2}$. Interestingly, mtDNA variant segregation was delayed in Δmrx6 cells and a higher percentage of cells remained heteroplasmic after 18 h of

◄ **Figure 5. Partitioning of a limited number of mtDNA molecules to the next generation facilitates rapid heteroplasmy changes.**

(A) Schematic of cell relationship pairs. Correlations were calculated between mother cells and their progeny within a lineage (M-D-GD) or between themselves at later timepoints (M-M$_D$-M$_{GD}$). All cells were taken at the same growth stage, specifically 20% bud-to-mother volume ratio. (B) The box plot depicts Spearman's correlation coefficient for the aforementioned relationship pairs, among all cell populations ($N = 9$). Each cell population was derived from an individual zygote. The Spearman's correlation coefficient for each population is shown as a gray dot. The box extends from the lower to upper quartile values of the data, with a line at the median. Whiskers indicate the minimum and maximum values. $P < 0.05$: *, $P < 0.005$: **, $P < 0.0005$: ***, paired $t$-test. (C) Representative time-lapse of a spot being transferred to the bud. Yeast cells expressing 3xNG-LacI (cyan) and matrix-su9-mKate2 (magenta) labeling nucleoids and the matrix, respectively, were imaged for 5 min in 10 s intervals. The white arrowhead indicates a 3xNG-LacI focus crossing the bud neck. Amounts of foci crossing the bud neck during 5-min windows were counted. Fluorescent images are maximum-intensity projections of z-stacks, after deconvolution. Scale bar: 5 µm. (D) Percentile proportions of mtDNA foci crossing the mother-bud neck in 5 min ($N = 93$ mother-bud pairs). Budding cells were at different cell cycle stages. Each spot was counted as one mtDNA copy. (E) The bar plot shows the time duration during which mitochondrial networks remain connected between mothers and daughters. Mitochondrial network connectivity was examined in log-phase cells expressing matrix-targeted mKate2 ($n = 33$). Only cells with no visible bud at the beginning of the imaging were considered for analysis (see also Appendix Fig. S8C). (F) The calculation formula for the number of nucleoids passing per cell division, based on data derived from (C) and (E). (G) Heteroplasmy distribution in Δ*mrx6* and Δ*dnm1* deletion strains. Cells with increased copy number (Δ*mrx6*) or deficient for mitochondrial fission (Δ*dnm1*) were generated in strains containing mtDNA$^{Atp6-NG}$ or mtDNA$^{Atp6-mKate2}$. Cells containing mtDNA$^{Atp6-NG}$ or mtDNA$^{Atp6-mKate2}$ and the respective deletion were mated. After zygote formation and dissection, diploid colonies were grown for 18 h on a YPD plate. Each colony was imaged to assess the fraction of heteroplasmic cells (also see Appendix Fig. S9A for all data points). Of note, Δ*dnm1* cells were cultivated in YPG media before mating and dissection on YPD plates, to prevent mtDNA loss. For each genotype shown, the small dots represent the percentage of heteroplasmic cells in individual colonies ($n > 50$ cells/colony). Each bigger dot depicts the mean of each biological replicate ($N = 5$), and the line is the mean of all replicates for the corresponding mating. Statistical significance was determined by paired $t$-test. (H) 24 h heteroplasmy assessment in matings between cells containing mutated mtDNA (Δ*cob*) or intact mtDNA. Log-phase yeast cells harboring mtDNA$^{Atp6-NG}$ were mated with cells containing intact intron-containing mtDNA, intact intronless mtDNA, or intronless Δ*cob* mtDNA, all not expressing any fluorophore. After zygote formation and dissection, diploid colonies were grown for 24 h on a YPD plate. Each colony was imaged to assess the fraction of cells not expressing Atp6-NG. For each genotype shown, the small dots represent the percentage of heteroplasmic cells in individual colonies ($n > 100$ cells/colony). Each bigger dot depicts the mean of each biological replicate ($N = 5$), and the line is the mean of all replicates for the corresponding mating. Error bars indicate SD. Statistical significance was determined by paired $t$-test. Source data are available online for this figure.

growth (Fig. 5G), with none of the populations having entirely segregated (Appendix Fig. S9B). In conjunction with predictions from our model (Fig. EV3), the experimentally determined delayed segregation in Δ*mrx6* cells suggests that these cells have higher fission-fusion frequencies and/or transmit a higher percentage of mtDNA copies to daughter cells.

Thus, our experimental data suggest that unequal amounts of mtDNA copies are partitioned to daughter cells during the cell cycle, which appears to be a major driving force for mtDNA segregation. Additionally, we find that fission deficiency leads to faster mtDNA variant segregation, while increased mtDNA copy number results in slower homoplasmy establishment.

Finally, we asked if we could apply our experimental setup to observe purifying selection of intact over mutant mtDNA. Currently, no yeast strains are available that harbor a mutant mtDNA encoding a fluorescent protein. Therefore, we conducted experiments in which cells containing intact intron-containing mtDNA$^{Atp6-NG}$ were mated with cells containing "dark" intronless mtDNA$^{il-\Delta cob}$, lacking the open reading frame encoding Cytochrome b, which is essential for respiratory growth. Notably, deletion of *COB* was generated in mtDNA lacking introns (Gruschke et al, 2011). Therefore, we conducted control experiments where cells containing mtDNA$^{Atp6-NG}$ were mated with cells harboring intact intron-containing (mtDNA$^{ic}$) or intact intronless mtDNA (mtDNA$^{il}$), both lacking genes encoding fluorescent proteins. Upon growth for 24 h, we assessed the percentage of "dark" cells, lacking mtDNA$^{Atp6-NG}$, in populations arising from heteroplasmic zygotes (Fig. EV5). Our data show that in matings between cells harboring mtDNA$^{Atp6-NG}$ and cells harboring mtDNA$^{ic}$, no preference for either mtDNA variant was present as equal segregation into 'dark' and fluorescent cells was observed (Fig. 5H). Matings between cells harboring mtDNA$^{Atp6-NG}$ and cells harboring mtDNA$^{il}$ revealed a slight preference for the intron-containing mtDNA as less "dark" cells were detected. Most strikingly, the number of "dark" cells was significantly lower after the growth of zygotes

containing mtDNA$^{Atp6-NG}$ and mtDNA$^{il-\Delta cob}$, indicating selection against the mutant mtDNA. This result confirms our previous finding that yeast cells favor generation of progeny with functional mtDNA copies (Jakubke et al, 2021), and that our experimental pipeline can be used to study the purifying selection of mtDNA.

## Discussion

In this study, we unveil the dynamics of mtDNA heteroplasmy in a living cell population with unprecedented temporal resolution. Our findings demonstrate that proliferating heteroplasmic yeast populations exhibit rapid transitions towards homoplasmic states. Through the analysis of heteroplasmic variance in cell lineages, alongside mathematical modeling and direct mtDNA copy tracking, we demonstrate that asymmetric partitioning of mtDNA copy numbers and mitochondrial fission are important determinants for mtDNA segregation. Our finding, with smaller daughter cells receiving a lower copy number than their mothers, also aligns with our previous finding that mtDNA copy number correlates strongly with cell volume (Seel et al, 2023).

Understanding the principles and dynamics that underlie the segregation of mtDNA variants has been hampered by the inability to track their presence and quantities within living cells across multiple cell divisions. We overcome this hurdle and infer the presence of heteroplasmy levels within individual cells over time, by employing two mtDNA variants encoding Atp6 tagged with different fluorescent proteins. Our results demonstrate the efficacy of this approach in assessing mtDNA quantities, despite the fact that proteins indirectly represent mtDNA. First, we observed a rapid decrease in fluorescent signals upon translational inhibition with Chloramphenicol and a subsequent resurgence of signal upon drug removal. Second, deletion of the mitochondrial HMG-box protein Abf2, causing gradual loss of mtDNA (Schrott and Osman, 2023; Sia et al, 2009; Zelenaya-Troitskaya et al, 1998), results in

heterogeneity of fluorescent signals within populations, whereas fluorescent signals in wild-type populations show little cell-to-cell variance. Hence, our fluorescent readout detects changes in mtDNA abundance or its expression within the duration of our experiments. These findings strongly affirm the notion that rapid changes in fluorescent intensities within proliferating cell populations, derived from heteroplasmic zygotes, closely align with shifts in the level of heteroplasmy. Nevertheless, it is very probable that residual "fossil" Atp6 protein amounts persist within cells, even if the mtDNA genomes encoding these proteins are not present in cells anymore, which likely leads to an underestimation of the speed of mtDNA variant loss.

Examination of mtDNA segregation dynamics has so far largely been limited to in silico approaches (Aryaman et al, 2019; Glastad and Johnston, 2023; Johnston, 2019; Johnston et al, 2015; Kowald and Kirkwood, 2011; Tam et al, 2013). As our experimental setup allows assessment of this process in vivo, comparison of empirical data with mathematical modeling becomes possible. To this end, we performed simulations, which consider important aspects imposed on mtDNA segregation by mitochondrial morphology and dynamics as well as the relaxed replication of mtDNA. Our model identifies shuffling and unequal partitioning as important factors influencing the speed of mtDNA variant segregation. Decreased shuffling of mtDNA increases the speed of segregation, likely because mtDNA copies stay close to their template mtDNA after replication and to other molecules copied from the same template. Thus, it is more probable that larger numbers of mtDNA copies of the same type are passed on to the daughter cell, which also implies that the nucleoids staying in the mother cell are more homogeneous. Indeed, our experiments confirm the prediction that lack of mitochondrial fission accelerates mtDNA variant segregation. This finding is also in line with previous mathematical modeling (Tam et al, 2013) and the reported limited mobility of mtDNA (Jakubke et al, 2021; Nunnari et al, 1997), which likely prevents shuffling of different mtDNA variants within the mitochondrial network in the absence of mitochondrial fusion and fission.

Shuffling and unequal partitioning are both predicted to influence mtDNA segregation. Direct assessment of mtDNA partitioning to progeny cells reveals that under our experimental conditions, about a third of available mtDNA copies are partitioned to the daughter cell in wild-type cells. Our model suggests that for such an *ndau* value, a relatively low *nspl* parameter (*nspl* = 5) is required to match the observed speed of mtDNA variant segregation. We would like to emphasize that the absolute value of the *nspl* parameter predicted by our model should be regarded with care. Our model does not consider parameters such as the branching of mitochondrial tubules, the spatial organization of mitochondrial tubules in the cell, or a link between mtDNA replication and mitochondrial fission (Lewis et al, 2016; Murley et al, 2013). Such parameters likely affect shuffling of mtDNA variants (Glastad and Johnston, 2023), and thereby the outcome of mtDNA segregation. Nevertheless, we regard it as an interesting possibility that the number of mtDNA molecules partitioned to the daughter cell, as well as fusion-fission cycles, can be modulated by the cell, perhaps in response to stress conditions or altered mitochondrial physiology to either promote or reduce the speed of mtDNA segregation.

Interestingly, we find that mtDNA variant segregation is slowed down in Δ*mrx6* cells that exhibit an increased mtDNA copy number. In light of our simulations, this result suggests that Δ*mrx6* cells do either pass on a higher percentage of mtDNA copies to daughter cells or that the fusion-fission frequency is increased. Given the link between mtDNA replication and mitochondrial fission (Lewis et al, 2016; Murley et al, 2013), it is an interesting possibility that increased mtDNA copy number and, consequently, more replication events, may cause an elevated fission frequency. It will be interesting to determine the cause for decreased mtDNA segregation speed in cells with elevated mtDNA content in future experiments.

In addition to assessing the segregation dynamics of two neutral mtDNA variants that fully support mitochondrial function, we employed our experimental setup to investigate the competition between intact and mutant mtDNA. Our findings corroborate our previous observations that *S. cerevisiae* cells preferentially generate progeny with intact mtDNA (Jakubke et al, 2021). Therefore, we anticipate that our approach will be instrumental in examining mtDNA quality control and purifying selection in future studies.

## Methods

### Yeast strain construction

All yeast strains employed in this study were generated in the W303 background. Detailed genotypes are listed in Appendix Table S1. Construction of single deletion mutants followed the procedure described in (Janke et al, 2004). Primers and plasmid information are listed in Appendix Tables S2 and S3, respectively. For the deletion of *ABF2*, transformed cells were plated and kept on non-fermentable (YPG) carbon source, to avoid loss of mtDNA prior to the microfluidic experiment.

### Yeast growth conditions

Prior to all experiments, cells were freshly streaked from −80 °C on YPD plates, left to grow at 30 °C overnight, and then inoculated in fermentable (YPD) liquid culture. The next day, cells were diluted to 0.1 $OD_{600}$ and further incubated to reach the log phase. Of note, the Δ*abf2* cells, for either variant, were maintained on YPG, to prevent spontaneous mtDNA loss.

### Growth assay and petite analysis

For the growth tests, log-phase cells were serially diluted and 3 µl of each dilution were spotted onto fermentable (YPD) and non-fermentable (YPG) plates. Plates were incubated at 30 and 37 °C, and were imaged after 24 and 48 h of growth. Appendix Fig. S1A shows the data after 48 h of growth. For the petite analyses to assess the loss of mtDNA in cells, 200 cells growing in log-phase were plated on YPG plates containing 0.1% glucose. Plates were imaged and colonies were counted after 3 days at 30 °C and classified into grande (big) or petite (small) colonies. Cells unable to properly respire show a petite phenotype.

## Live cell microscopy

Time-lapse data was acquired using the microfluidics CellAsic ONIX system (Millipore) attached to a Nikon Eclipse Ti inverted microscope, equipped with a CFI Apochromat TIRF 100XC oil objective with a numerical aperture of 1.49 and a working distance of 0.12 mm (Nikon MRD01991). A TwinCam LS dual-camera splitter, coupled with two Photometrics Prime 95B 25-mm cameras, facilitated simultaneous imaging of red and green fluorophores. Every experiment was performed under the same settings for both fluorophores, with 40 ms exposure time, and 40% LED power. NeonGreen was excited at 506 nm wavelength, while mKate2 at 588 nm. Z-stacks were captured with increments of 0.3 μm over a total of 6 μm. Information on filters and dichroics can be made available upon request.

Both haploid and diploid cells were visualized for 8 up to 12 h, depending on the experiment, and images were acquired at 30 °C in 15 min intervals throughout the duration of the time-lapse using a temperature-controlled chamber around the microscope. Cells were continuously replenished with a fresh minimal medium at a flow rate of 15 kPa. For all microfluidic experiments, the CellAsic ONIX plates (Millipore) for haploid yeast cells were used. To obtain the diploid cells used for diploid live cell imaging, log-phase haploid cells harboring Atp6-NG were mated with haploid Atp6-mKate2 cells, in a 1:1 ratio on a YPD plate. After 2 h at 30 °C, once zygotes were formed, cells were resuspended in minimal medium, and single zygotes were picked in individual field of views to be monitored overnight using the CellAsic ONIX system.

## Image processing and data acquisition

Cell segmentation and tracking was performed frame by frame based on the DIC channel using the Cell-ACDC software and the implemented YeastMate cell segmentation algorithm (Bunk et al, 2022; Padovani et al, 2022). Cell cycle annotations, mask correction, and lineage tracking errors, as well as cell volume calculation, were also performed through the Cell-ACDC software. All cell masks and traces were also manually verified. Fluorescent signals were quantified from the mean projection of the z-stack per timepoint from non-deconvolved video recordings. Fluorescent intensities per cell were first corrected by subtraction of background signal, measured in cell-free areas, and normalized to cell volume. Subsequently, for comparability between the different cell populations and to account for photobleaching, the cell signal intensity for the respective fluorophore was additionally normalized to the median intensity of all segmented cells of the given timepoint and technical replicate. Microscopy images in all figures were deconvolved with the Huygens software (Scientific Volume Imaging).

## 24-h heteroplasmy assessment

For assessing the proportion of homoplasmic Atp6-mKate2 and Atp6-NG cells after 24 h, the respective strains were grown to log phase and mated in a 1:1 ratio for 90 min on YPD plates. Once zygotes were formed, they were micro-dissected and placed in separate areas on a YPD plate, where they grew into single colonies overnight (Fig. EV2). The next morning, each distinct colony was picked and diluted into 200 μl and imaged on an Ibidi glass slide, previously coated with the immobilizing agent Concanavalin A (Jakubke et al, 2021). To get an estimate of heteroplasmy within the population, a minimum of 200 cells were imaged and evaluated from each individual colony. First, the h-value of all segmented cells was calculated, using the heteroplasmy equation (Johnston et al, 2015), and subsequently, after binning the data, cells were characterized as homoplasmic for Atp6-NG, Atp6-mKate2 or heteroplasmic if they fell within the middle range of heteroplasmy, defined by the intersection points of the single Gaussian curves (Fig. 3C).

For assessing the level of homoplasmy in cells competing between intact and mutant mtDNA, we mated WT strains harboring mtDNA$^{Atp6-NG}$ with cells containing intronless mtDNA$^{\Delta cob}$ (Gruschke et al, 2011). As a control, we also mated cells harboring mtDNA$^{Atp6-NG}$ with WT "dark" mtDNA$^{ic}$, or WT "dark" mtDNA$^{il}$. To assess the percentage of homoplasmic cells, we manually determined a threshold for 'signal' or 'no signal' by thresholding the fluorescent intensity of Atp6-NG based on the distribution of cells across each replicate as follows. Across all experiments, two distributions, exhibiting or virtually lacking fluorescent signals, were observed. The lowest value at the valley between the two peaks was set as the threshold for signal per replicate.

For assessing the heteroplasmy levels of cells with increased copy number (Δmrx6) or fission-deficient ones (Δdnm1), the zygote dissection pipeline was performed as described above, only that colonies were imaged after 18 h of growth on YPD plates. The rationale for earlier imaging was to capture differences in the percentage of remaining heteroplasmic cells more clearly compared to later imaging, where mtDNA variant segregation would have been virtually complete even in matings with a decreased rate of segregation.

## mtDNA copy partitioning during cell division

To estimate the number of mtDNA copies crossing the mother-bud neck, we used a diploid strain developed in the lab, containing mtDNA with integrated LacO repeats and expressing the nuclear-encoded matrix-targeted 3xNG-LacI protein as well as a matrix-targeted mKate2 to monitor the mitochondrial network (3xNG-LacI-PGK1-su9-mKate2) (Osman et al, 2015). Log-phase cells, pre-grown in YPD, were visualized in 10-second intervals for 5 min in total, since longer exposure (with such frequent intervals) induces significant photobleaching.

By focusing exclusively on the neck region where the two cells are joined, we quantified the migration of mtDNA fluorescent foci from the maternal to the daughter side. Notably, we only analyzed cells where the mitochondrial networks between mother and bud were connected. Given that cells were at different stages of the cell cycle, we calculated the average amount of foci transmitted in 5 min, regardless of bud size. We cannot rule out that if two mtDNA foci are in close proximity, they might be counted as a single molecule. Of note, all aforementioned analysis was done using all z-slices and 3D modeling of each cell, and no maximum projection of z-stacks, to avoid loss or misinterpretation of the number of foci per cell pair.

To precisely determine the duration of content exchange between mothers and daughters, we employed the same strain but under different imaging conditions. Log-phase cells were

visualized for 120 min, in 5-min intervals, for successful monitoring of complete division cycles and mitochondrial network association between mothers and daughters. We defined the initiation of budding, when cells were in the very early S phase, as timepoint zero. We tracked the exchange process until the daughter's mitochondrial network had completely separated from that of the mother. At this point, just before the ultimate separation of networks, most daughter cells had developed into buds with an average cell volume equivalent to 55% of the mother's size, most likely corresponding to mid- to late G2 phase. In cases where mothers had large buds, we observed no network connectivity, indicating that these buds had already entered the late G2 phase in preparation for cytokinesis, which is consistent with existing literature (Koren et al, 2010). By analyzing 33 cells from timepoint zero to the last timepoint, we calculated an average mitochondrial content exchange period of 47.5 min, which was subsequently used to calculate an estimate of nucleoids transmitted per division.

### mtDNA foci assessment during the cell cycle

In order to assess the number of mtDNA foci during a complete cell cycle, we used the same diploid strain (3xNG-LacI-PGK1-su9-mKate2) used for mtDNA foci assessment during mother-bud partitioning. Cells were harvested at log phase and imaged for 2 h at 7 min intervals. mKate2 and NeonGreen were simultaneously captured using the dual-camera system (50 ms exposure time, 30 and 100% light intensity, respectively). Of note, all aforementioned analyses were done using all z-slices and 3D modeling of each cell, and no maximum projection of z-stacks, to avoid loss or misinterpretation of the number of foci per cell pair. Foci were counted with custom-built Fiji Macros using the "3D Maxima finder" command. For the analysis, only virgin mothers, that is, cells that had just budded off from their mother, were considered. Timepoint zero was determined based on the point of bud emergence from the virgin mothers, per time-lapse video.

### Mathematical model

In the model, we represent mitochondria as lists of 0 and 1 s, representing two different mtDNA genotypes. Mitochondria are modeled as one-dimensional objects without branches. When a mtDNA molecule is duplicated, the copy is placed right next to the original one. For the mtDNA copy number of diploid yeast cells, we apply actual measurements derived from (Göke et al, 2020), and set the average amount of mtDNA copies to 32 in each simulated cell. We additionally assume that one nucleoid contains one mtDNA copy and all mtDNA copies have identical replication rates. Importantly, the model neglects the possibility of a preferential replication of only a fixed subset of copies at any timepoint.

In order to accurately reflect the heteroplasmic state of founder cells, we define two different mtDNA variants in each zygote, mtDNA type 0 and mtDNA type 1. Hence, a cell can have two different states, homoplasmic for either 0 or 1, or a combination of both when heteroplasmic. In each simulation, founder cells were consistently characterized by virtually absolute heteroplasmy, featuring a distribution of mtDNA haplotypes according to the average of all nine biological replicates and a mitochondrion structure composed of alternating 0 and 1 sequences (that is 0101010101… in the case of 50% mtDNA haplotype 1). Given our

experimental observations, where spontaneous mtDNA depletion is absent, we also postulate the absence of mtDNA depletion in each cell in silico. Furthermore, to simulate yeast lineages as close to the real data as possible, we allow each cell to divide only upon reaching the predefined mtDNA copy number. This way, when a daughter cell is created, it is allowed to first grow and replicate its content, before proceeding to division.

To account for the fission and fusion events in mitochondria, we establish the *nspl* parameter, defining the number of fragments every mitochondrial sequence splits into. This fragmentation leads to a reshuffling of the mtDNA copies within each mitochondrion, mimicking the fission-fusion dynamics. Upon this shuffling process, all fragments refuse and each cell is allowed to divide, by splitting into two parts. We also define a second variable *ndau* for the number of copies a mother cell gives to the daughter per division. We set the maximum *ndau* value to always be half of the total number of copies in the given founder cell. For example, in the simulations with increased copy number, where $n = 56$ or $n = 90$ copies/cell, each daughter cell could inherit a maximum of 28 or 45 copies, respectively. The *ndau* parameter, along with the *nspl*, are the only model parameters to be fitted to the data. Specifically, we first fit a sigmoid curve to the experimental heteroplasmy values at six timepoints ($t = 1.5, 3, 4.5, 6, 7.5$ h) and then identified the best-fitting pairs (*ndau*, *nspl*), using least-squares fitting.

### Data analysis

Data analysis and statistics were performed in Python and the in silico mathematical modeling was constructed using R. A dependent two-sided *t*-test (scipy.stats.ttest_rel) was chosen for related lineage data (Fig. 5B). An independent two-sided *t*-test (scipy.stats.ttest_ind) was chosen for all data derived from independent samples (Fig. 5G,H). All scripts are provided in a GitHub repository under https://github.com/statgenlmu/yeast_mito_segregation.

## Data availability

This study includes no data deposited in external repositories. Scripts and code for data analysis are deposited in the GitHub repository under https://github.com/statgenlmu/yeast_mito_segregation.

The source data of this paper are collected in the following database record: biostudies:S-SCDT-10_1038-S44318-024-00183-5.

## Peer review information

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

## Acknowledgements

We thank Nadja Lebedeva and Johannes Hagen for their technical assistance. Appreciation is extended to all members of the Osman and Mokranjac groups for providing constructive feedback. Furthermore, we are grateful for the enriching discussions within the "Mito-Club". This work was funded by a European Research Council Starting Grant (ERCStG-714739 IlluMitoDNA), a Human Frontier Science Program Research Grant (RGP021/2023) and a DFG Research Grant (HO 7333/1).

## Author contributions

**Rodaria Roussou:** Conceptualization; Data curation; Formal analysis; Investigation; Methodology; Writing—original draft; Writing—review and editing. **Dirk Metzler:** Conceptualization; Software; Formal analysis; Writing—original draft; Writing—review and editing. **Francesco Padovani:** Software; Writing—review and editing. **Felix Thoma:** Formal analysis; Investigation; Methodology. **Rebecca Schwarz:** Formal analysis; Investigation. **Boris Shraiman:** Conceptualization. **Kurt M Schmoller:** Conceptualization; Software; Supervision; Funding acquisition; Writing—original draft; Writing—review and editing. **Christof Osman:** Conceptualization; Resources; Supervision; Funding acquisition; Methodology; Writing—original draft; Project administration; Writing—review and editing.

Source data underlying figure panels in this paper may have individual authorship assigned. Where available, figure panel/source data authorship is listed in the following database record: biostudies:S-SCDT-10_1038-S44318-024-00183-5.

## Funding

## Disclosure and competing interests statement

The authors declare no competing interests.

# Expanded View Figures

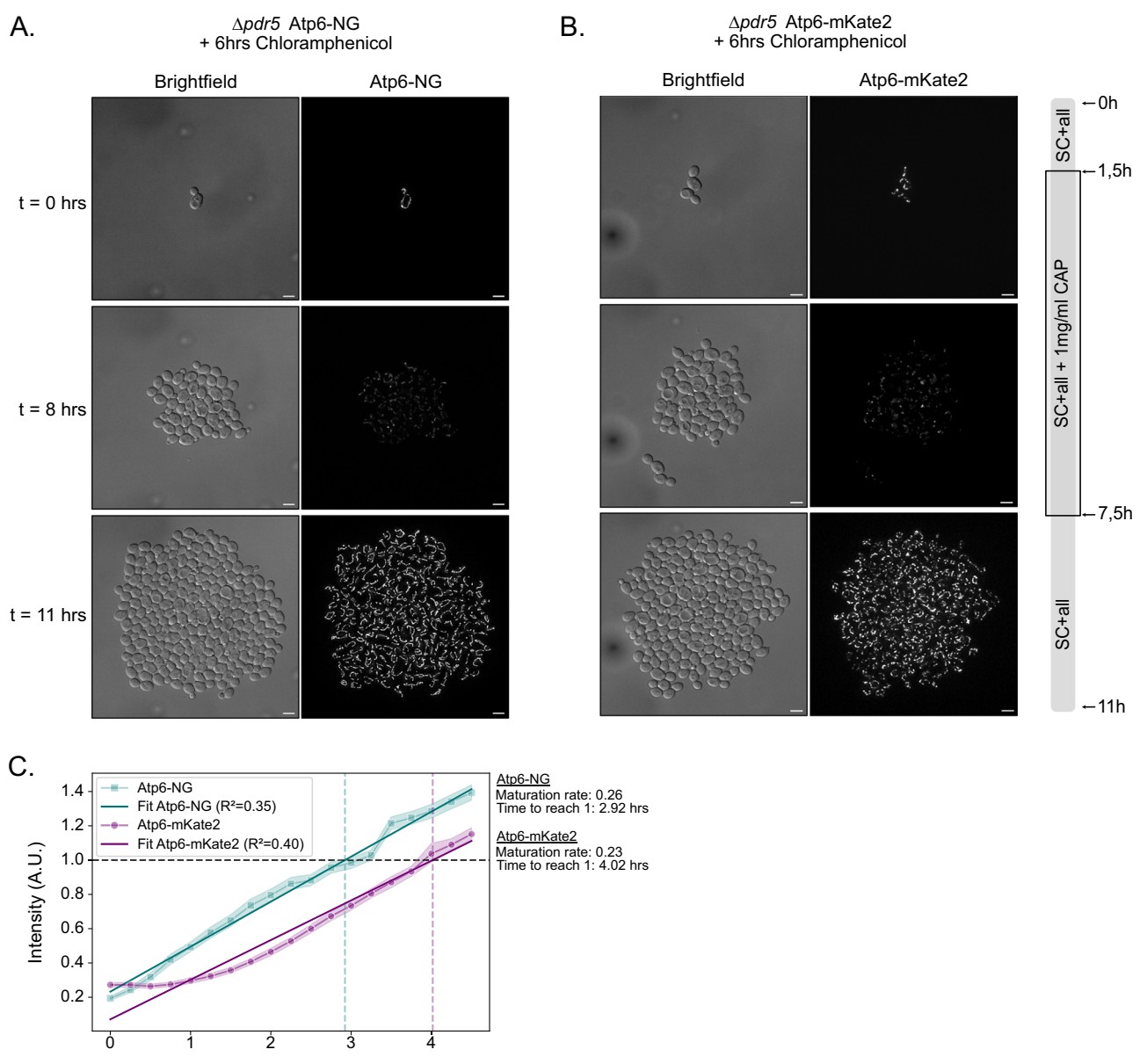

**Figure EV1. mtDNA^Atp6-NG and mtDNA^Atp6-mKate2 strains with temporary exposure to chloramphenicol.**

(A, B) Yeast cells harboring mtDNA^Atp6-NG (A) or mtDNA^Atp6-mKate2 (B) were imaged for 11 h (N = 3/strain). Chloramphenicol (1 mg/ml) was added after the first cell duplication, i.e., after 1.5 h of imaging, and was removed from the microfluidic chamber after 6 h of incubation. Fluorescence images are maximum-intensity projections of z-stacks, after deconvolution. Scale bars: 5 μm. (C) Line graph showing the fluorescent intensities of cells expressing Atp6-NG (cyan) or Atp6-mKate2 (magenta) upon washing off the Chloramphenicol (see Fig. 2F) after incubation with 1 mg/ml CAP for 6 h for sufficient mitochondrial translation inhibition. Curve fitting on the fluorescent intensity lines upon washing off the drug provides the maturation timings of the two Atp6 variants. Data represent the mean of all replicates per strain, and the shaded areas represent the 95% confidence interval. Fluorescent intensities were normalized to the cells at the first timepoint, for either of the fluorescent channels.

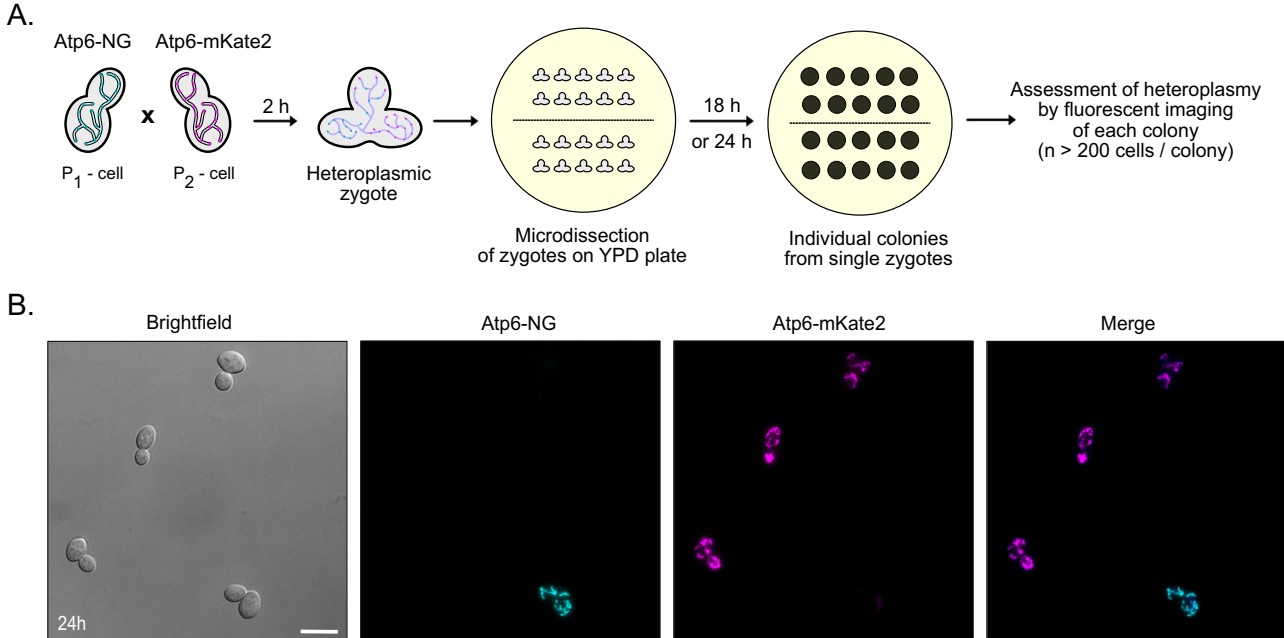

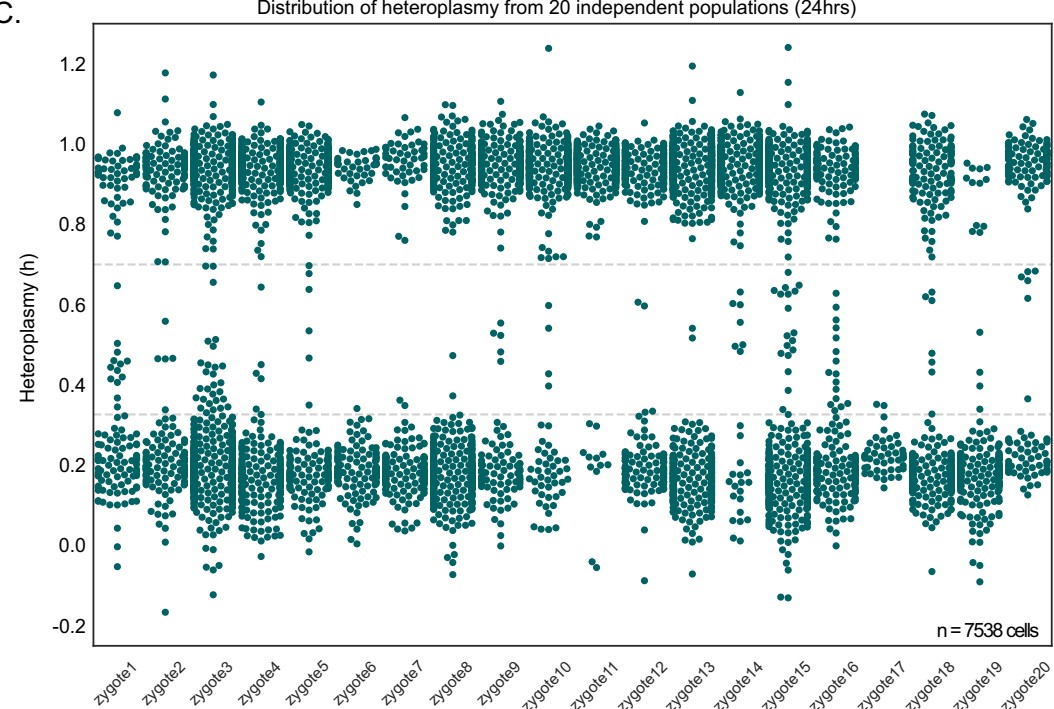

**Figure EV2. 24-h heteroplasmy assessment from 20 independent populations.**

(A) Schematic of the 24-h heteroplasmy assessment experiment. Cells harboring mtDNA$^{Atp6\text{-}NG}$ were mated with cells containing mtDNA$^{Atp6\text{-}mKate2}$ on a YPD plate. After a 2-h incubation, upon zygote formation, individual zygotes were micro-dissected and placed in different areas of a fresh YPD plate. Cells were kept for 24 h at 30 °C until colonies were formed. Cells from each colony, originating from a single zygote, were imaged to assess population homoplasmy levels, by calculating the proportion of cells exhibiting Atp6-NG and/or Atp6-mKate2 signal. (B) Representative image of a small field of view from one population, after 24 h. In total, 20 individual populations were screened, each originating from a diploid heteroplasmic zygote. Cells harboring mtDNA$^{Atp6\text{-}NG}$ are shown in cyan, while cells having kept the mtDNA$^{Atp6\text{-}mKate2}$ are depicted in magenta. Fluorescence images are maximum-intensity projections of z-stacks, after deconvolution. Scale bars: 5 µm. (C) Heteroplasmy distributions from all 20 independent diploid populations after 24 h of growth on the YPD plate ($N = 20$ zygotes/$n > 200$ cells/colony). Each dot represents a single cell. Heteroplasmy values below 0.33 or above 0.71, the homoplasmy thresholds, represent cells harboring mainly mtDNA$^{Atp6\text{-}NG}$ or mtDNA$^{Atp6\text{-}mKate2}$, respectively. As apparent from the graph, some cells for each population still exhibit a heteroplasmic state, which is evident from a h-value between 0.33 and 0.71.

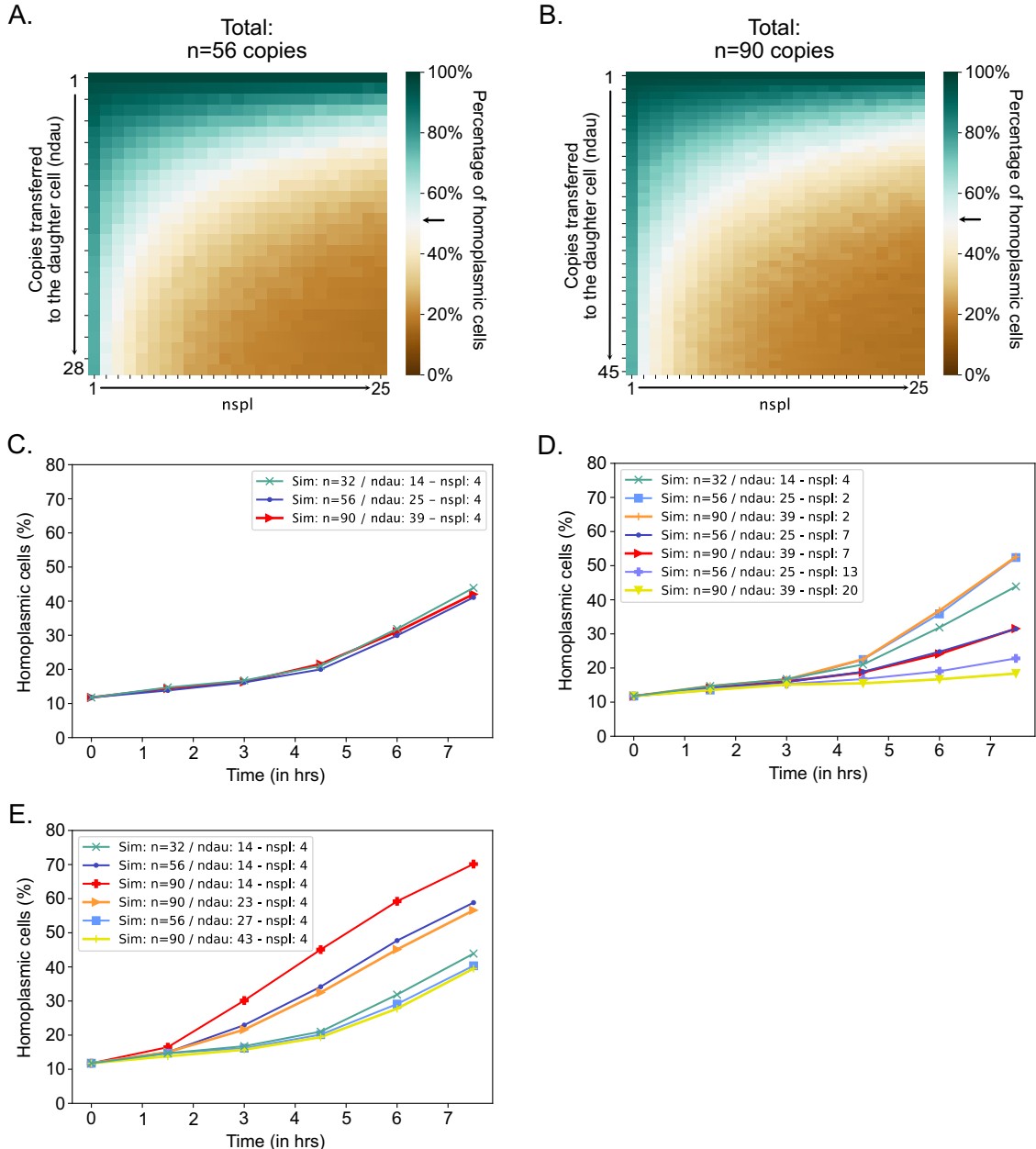

**Figure EV3. Simulations of homoplasmy establishment in cell populations with increased mtDNA copy number.**

(A) Heatmap showing the percentage of homoplasmic cells at timepoint $t = 7.5$ h, for different *ndau* and *nspl* pairs, for $n = 56$ mtDNA copies. The maximum value of *ndau* is defined by the half ($n = 28$) of the total mtDNA copies in the founder cell ($n = 56$). The arrow indicates the proportion (50%) of cells being virtually homoplasmic in the empirical data, based on the pre-established homoplasmy cutoffs. Each simulation with any given combination of the two parameters has been run ten times. (B) Heatmap showing the percentage of homoplasmic cells at timepoint $t = 7.5$ h, for different *ndau* and *nspl* pairs, for $n = 90$ mtDNA copies. The maximum value of *ndau* is defined by half ($n = 45$) of the total mtDNA copies in the founder cell ($n = 90$). The arrow indicates the proportion (50%) of cells being virtually homoplasmic in the empirical data, based on the pre-established homoplasmy cutoffs. Each simulation with any given combination of the two parameters has been run ten times. (C) Percentage of homoplasmic cells in simulated cell populations with $n = 32$ copies, relative to populations with $n = 56$ copies or $n = 90$ copies/cell. The number of copies transferred to the daughter cell (*ndau*) has been kept proportional to the total copy number in each of the simulations, that is 44\% of the total copies gets transferred to the daughter cell. The parameter *nspl* has been kept identical. (D) Percentage of homoplasmic cells in simulated cell populations with $n = 32$ copies, relative to populations with $n = 56$ copies or $n = 90$ copies/cell, with varying *nspl* values. The *ndau* parameter is proportional to the total copy number (44%), while the *nspl* parameter varies from low to high values per simulation. A comparison of these curves demonstrates the effect of lower or higher fission-fusion frequencies on the speed of segregation. The combination of *ndau* = 14 and *nspl* = 4 in cells with $n = 32$ copies is used as a point of reference, across all plots. (E) Percentage of homoplasmic cells in simulated cell populations with $n = 32$ copies, relative to populations with $n = 56$ copies or $n = 90$ copies/cell, with identical *nspl* values, and different levels of *ndau*. Comparison of these curves demonstrates the effect of lower or higher numbers of mtDNA copies being transferred to the daughter cells on the speed of segregation. The combination of *ndau* = 14 and *nspl* = 4 in cells with $n = 32$ copies is used as a point of reference, across all plots.

A.

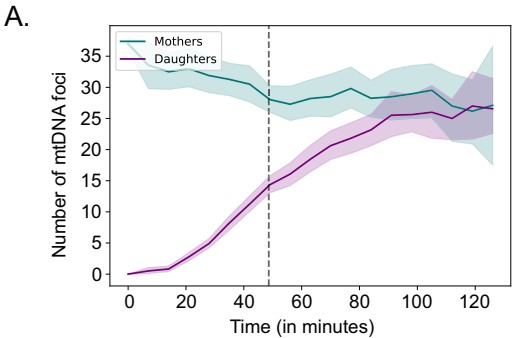

B.

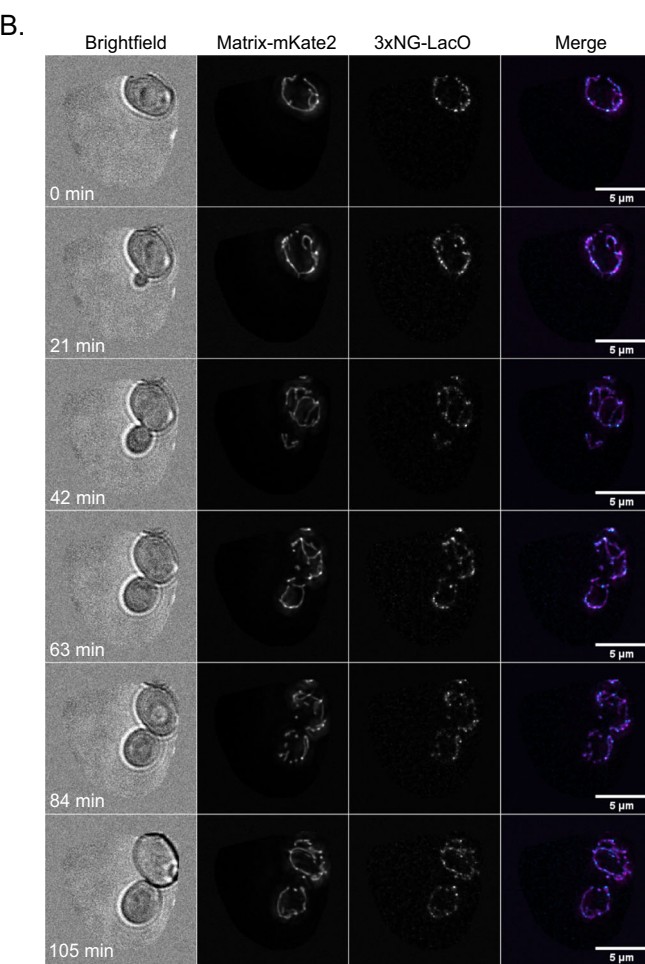

**Figure EV4. Number of mtDNA foci in mother-daughter pairs during a complete cell cycle.**

(A) Diploid cells expressing mitochondrially targeted 3xNG-LacI and mKate2 were harvested at the log phase and imaged for 2 h. The number of mtDNA foci were counted in virgin mothers (M) and their emerging daughters (D) ($n = 59$ M-D pairs). The dashed line represents the average period of time ($t = 47.5$ min) that mitochondrial content was exchanged between mothers and daughters, as shown in Fig. 5D. (B) Example time-lapse images of a virgin mother and its emerging bud. All cells express mitochondrially targeted 3xNG-LacI, used for visualizing the LacO-mtDNA foci, and mitochondrially targeted mKate2 to visualize the mitochondrial network. Mother-daughter pairs were cropped manually prior to segmentation and analysis.

A.

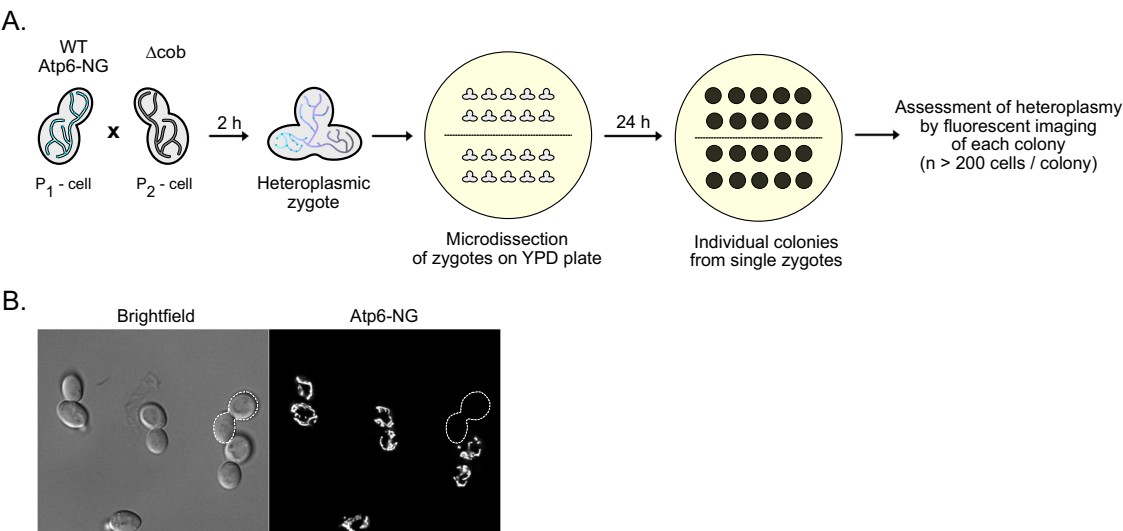

B.

Figure EV5.  24-hr heteroplasmy assessment in colonies with competing mtDNA qualities.

(A) Schematic of the 24 h heteroplasmy assessment experiment of matings between cells harboring intact or mutant mtDNA. Cells harboring intron-containing mtDNA$^{\text{Atp6-NG}}$ were mated with cells containing intronless mtDNA$^{il-\Delta cob}$ on a YPD plate. After a 2 h incubation, upon zygote formation, individual zygotes were micro-dissected and placed in different areas of a fresh YPD plate. Cells were kept for 24 h at 30 °C until colonies had formed. Cells from each colony, originating from a single zygote, were imaged to assess the percentage of cells with fluorescent (mtDNA$^{\text{Atp6-NG}}$) or 'dark' (mtDNA$^{il-\Delta cob}$) mitochondria. (B) Representative image of a small field-of-view from one population, after 24 h. Diploid cells harboring mtDNA$^{\text{Atp6-NG}}$ are exhibiting fluorescence, while 'dark' cells (indicated by a white outline) do not contain mtDNA$^{\text{Atp6-NG}}$. Fluorescence images are maximum-intensity projections of z-stacks, after deconvolution. Scale bars: 5 μm.

