## [Peer Review File · The EMBO Journal]

Real-time assessment of mitochondrial DNA heteroplasmy dynamics at the single-cell level

Rodaria Roussou, Dirk Metzler, Francesco Padovani, Felix Thoma, Rebecca Schwarz, Boris Shraiman, Kurt Schmoller, and Christof Osman

Corresponding author(s): Christof Osman (osman@bio.lmu.de)

Review Timeline:

Submission Date:	7th Mar 24
Editorial Decision:	8th Apr 24
Revision Received:	7th Jun 24
Accepted:	17th Jul 24

Editor: Hartmut Vodermaier

Transaction Report:

Prof. Christof Osman
Ludwig-Maximilians-Universitaet Muenchen
Faculty of Biology
Grosshaderner Str. 2
Planegg-Martinsried, Bayern 82152
Germany

8th Apr 2024

Re: EMBOJ-2024-117200
Real-time assessment of mitochondrial DNA heteroplasmy dynamics at the single-cell level

Dear Christof,

Thank you again for submitting your study on single-cell monitoring of mtDNA heteroplasmy dynamics to The EMBO Journal. I have now heard back from three expert referees, who had agreed to review the manuscript. As you will see, the reviewers appreciate your approaches and the importance of the question. At the same time, they all feel that the study would be considerably stronger if the new system would have been exploited to obtain deeper new insights into mtDNA inheritance/segregation beyond confirming previously suggested mechanisms. This is most explicitly stated in the major concern of referee 1, the general comment of referee 2, as well as in the more specific queries raised by referee 3.

Should you be able to deepen this aspect of the work, and to satisfactorily address the various specific/minor points listed in the reports, we would be happy to consider an adequately revised manuscript further for publication. Since our single-major-revision-round policy makes it important to diligently respond to each referee point at the time of resubmission, I would however encourage you to contact me with a preliminary point-by-point response already during the early stages of your revision work, in order to clarify how key issues may be addressed and to discuss possible revision plans (happily via Zoom if needed). We would also be open to extension of the default three-months revision period if needed; our 'scooping protection' (meaning that competing work appearing elsewhere in the meantime will not affect our considerations of your study) would of course remain valid also throughout such an extension.

Detailed information on preparing, formatting and uploading a revised manuscript can be found below and in our Guide to Authors. Thank you again for the opportunity to consider this work for The EMBO Journal, and I look forward to hearing from you in due time.

With kind regards,

Hartmut

9) Digital image enhancement is acceptable practice, as long as it accurately represents the original data and conforms to community standards. If a figure has been subjected to significant electronic manipulation, this must be clearly noted in the figure legend and/or the 'Materials and Methods' section. The editors reserve the right to request original versions of figures and the original images that were used to assemble the figure. Finally, we generally encourage uploading of numerical as well as gel/blot image source data; for details see: embopress.org/page/journal/14602075/authorguide#sourcedata

At EMBO Press, we ask authors to provide source data for the main manuscript figures. Our source data coordinator will contact you to discuss which figure panels we would need source data for and will also provide you with helpful tips on how to upload and organize the files.

In the interest of ensuring the conceptual advance provided by the work, we recommend submitting a revision within 3 months (7th Jul 2024). Please discuss the revision progress ahead of this time with the editor if you require more time to complete the revisions. Use the link below to submit your revision:

Link Not Available

Referee #1:

In this elegant manuscript, Roussou et al explore how heteroplasmy changes across generations in yeast lineages. Taking advantage of heteroplasmic yeast strains expressing differentially mitochondrially-encoded fluorescent proteins, they find that asymmetric partitioning of mtDNA during cell division drives mtDNA variant segregation. This study is of the highest rigor and will be of interest to a broad readership.

My one major comment has to do with further experimentally testing the main conclusion of the manuscript, viz. that the transmission of a limited number of mtDNA copies is a major driver for the progressive divergence of heteroplasmic states in a proliferating yeast population. I wonder if this conclusion could be further empirically tested for example by increasing copy number (TFAM overexpression) and determining if this reduces the rate of variant segregation. I don't think this experiment is absolutely necessary, but a further testing of their main finding would potentially strengthen the manuscript.

I have no other major comments. This manuscript was a pleasure to read and, I believe, a strong candidate for the EMBO J.

Minor Comments

- Should page 6 line 3 be 8 hours, not 6 hours?

- Using the mtLacO-LacI system they determine ~11 nucleoids migrate from mother to daughter. Assuming one genome per nucleoid, what can be inferred about the contribution of shuffling due to fission fusion cycles based on the author's modelling?

Referee #2:

In this study, Roussou et al have developed a novel approach which allows them to follow fluorescently labeled mtDNA variants in heteroplasmic yeast populations over several generations and with single-cell resolution. They show that mtDNA variant segregation is driven by asymmetric partitioning of mtDNA copies during cell division. Using computational modeling they conclude that these segregation kinetics are further influenced by mitochondrial fission and fusion.

The method and modeling developed for this manuscript represents a powerful method to investigate mtDNA inheritance. They could confirm previously suggested mechanisms about mtDNA variant segregation and the role of mitochondrial fission and fusion in this process. While these are important findings, the authors could have used this approach to explore further aspects of mtDNA inheritance, such as tracking the dynamics of mutated versus wildtype mtDNA. This would in my view significantly increase novelty and the impact of the study.

Major concerns:

1. From the description in the main text and methods section it is not clear if/how the authors have validated the automated tracking of cell over multiple generations. Where at least some of the data also manually annotated and resulted in the same lineage tree?
2. What is visualized is actually not the mtDNA itself, but rather the expressed protein (and its redistribution to the progenies). Although the authors perform experiments to show that the decay time for both Atp6-NG and Atp6-mKaede2 are similar, the half-life time seems to be in the range of hours. Therefore the loss of heteroplasmy might be significantly underestimated. Could this be corrected for in their model?
3. It is not clear to me why the wildtype mtDNA-Atp6-NG strain increases fluorescence over time, which the authors attribute to the switch in media, while the wildtype mtDNA-Atp6mKaede2 does not show this increase upon media switch (and even display a decrease in signal)?
4. When estimating the percentage of hetero- and homoplasmic populations (Fig. 3), autofluorescence in the "absent" channel is given as a potential explanation. It is surprising that such high levels of autofluorescence appear in healthy cells. I would suggest to compare the levels measured with wildtype yeast, which do not express any fluorescent protein, to verify if equally high levels of autofluorescence are measured.
5. Regarding the effect of fission and fusion on mtDNA inheritance, it has been shown previously, that mitochondrial fission is also linked with mtDNA replication. Is this accounted for in the model (and if not, why do the authors think it is neglectable)?

Referee #3:

mtDNA are present in multiple copies in a single cell. These can vary in sequence due to accumulation of mutations, resulting in heteroplasmy. The degree of heteroplasmy and the transition towards homoplasmy can have effects on mitochondrial function (dependent on the nature of the mtDNA mutations). The lack of available tools to visualize variant mtDNA copies within single cells has challenged the ability to track heteroplasmic mtDNA segregation dynamics. In this study, Roussou et al., use differently-labelled mitochondrial genomes to track mtDNA segregation and heteroplasmy in live cells. They first make homoplasmic ATP6 gene tagged with either mneongreen or mkate2. They then generate a heteroplasmic zygote expressing both variants together and track via live cell imaging the segregation of mkate2 and mneongreen across 6 generations of growth. They find that there is variation in the degree of heteroplasmy as cells divide, and over time cells move towards homoplasmy (with either mneongreen or mkate2 alone). They develop a model for these dynamics. They suggest a role for mitochondrial fission-fusion combined with asymmetric segregation of mtDNA between mother and daughter in mtDNA segregation.

Overall, the study is very interesting and provides insights into outstanding and important question in the field. The tool developed to track mtDNA in single cells is powerful and has significant future applications. I have a few comments/ suggestions that should further strengthen the manuscript:

1. Is there mixing of mtDNA variants? The transition from heteroplasmy to homoplasmy in the population is rapid. Given this timeframe, it is possible that the fluorescence is free diffusing across the network, but the mtDNA (nucleoid) variants in the zygote do not mix. Authors should assess whether there is mixing, or if the mtDNA (nucleoids) remain spatially segregated.
2. If there is no DNA mixing, is the present experimental setup truly reflective of heteroplasmy (as one would expect in physiological conditions)? A scenario where there is no mtDNA mixing could also result population transition towards homoplasmy, as at each division, cells would predominantly inherit only one of the two DNA variants.
3. Following on the above comment, does the model assume that there is mixing between the mtDNA variants in the zygote?

How would the segregation dynamics play out in a scenario where this did not occur?

4. The section on asymmetric partitioning needs further explanation/ analysis: For the asymmetric partitioning model to apply, daughter cells would have to inherit a fixed amount of mitochondria and mtDNA from the mother cell, early during bud formation. After this, as the daughter cell grows, mitochondria and mtDNA would have to be synthesized in the daughter cell. It is not so clear whether this is indeed the case, or that transmission of mitochondria and mtDNA occurs through the growth period of the bud up until division. Could the authors clarify this, and provide support for the scenario they are favouring? Authors should provide quantification of the number of nucleoids and cell area for mother and daughter cells at the time of division (when daughter and mother cell sizes should be comparable), and the number of nucleoids across cell area for mother and daughter cells from birth till division.

Minor point:

Page 6, second paragraph line 6: the figure reference appears to be incorrect.

Prof. Dr. Christof Osman · Großhaderner Str. 2 · 82131 Planegg-Martinsried

Dr. Hartmut Vodermaier

Prof. Dr. Christof Osman

Ludwig-Maximilians-Universität
Biocenter of the LMU
Großhaderner Str. 2
82152 Planegg-Martinsried

☎ (+49) 89-2180-74756

✉ osman@bio.lmu.de

June 6, 2024

Response to reviewer comments regarding our manuscript entitled: 'Real-time assessment of mitochondrial DNA heteroplasmy dynamics at the single-cell level'

Dear Hartmut,

We are delighted to receive positive feedback on our manuscript titled "Real-time assessment of mitochondrial DNA heteroplasmy dynamics at the single-cell level". We appreciate the time and effort you and the reviewers have dedicated to evaluating our submission. We are grateful for the comments and we believe that our additional work inspired by the comments has greatly improved the manuscript. Most importantly, we have included several new experiments to provide more insight into the biology that underlies mtDNA variant segregation. The most important additions are the following:

- We have experimentally assessed the impact of mitochondrial fission on mtDNA variant segregation. In line with predictions derived from our simulations, we find that deletion of the gene *DNM1* results in faster segregation of mtDNA variants.
- We have included a computational and experimental assessment of the impact of higher mtDNA copy number on the segregation of mtDNA variants. We find that a strain with increased mtDNA copy number exhibits a slower segregation of mtDNA variants. We discuss this finding in light of predictions made by our simulations.
- We demonstrate that our experimental pipeline can also be used to examine segregation of WT and mutant mtDNA by observing purifying selection against mutant mtDNA.

Reviewer #1:

My one major comment has to do with further experimentally testing the main conclusion of the manuscript, viz. that the transmission of a limited number of mtDNA copies is a major driver for

the progressive divergence of heteroplasmic states in a proliferating yeast population. I wonder if this conclusion could be further empirically tested for example by increasing copy number (TFAM overexpression) and determining if this reduces the rate of variant segregation. I don't think this experiment is absolutely necessary, but a further testing of their main finding would potentially strengthen the manuscript.

We are happy to hear that the reviewer liked our manuscript and are grateful for the suggestion of an experiment to test whether increasing mtDNA copy number reduces the rate of variant segregation. In our revised manuscript, we have examined the effect of increased mtDNA copy number on mtDNA variant segregation by simulations and also experimentally by examining segregation in cells lacking the gene *MRX6*, which was previously shown to exhibit a twofold increase of mtDNA copy number (Göke et al., MBoC, 2020). As hypothesized by the reviewer, we indeed see a reduced rate of variant segregation in $\Delta mrx6$ cells. Based on predictions made by our simulations, we discuss that the reduced rate of variant segregation is caused by a higher percentage of mtDNA copies passed on to daughter cells, an increased mitochondrial fusion and fission frequency or a combination of both. The results are now presented in Fig. 5G and S10.

Minor Comments

Should page 6 line 3 be 8 hours, not 6 hours?

This has been corrected.

Using the mtLacO-LacI system they determine 11 nucleoids migrate from mother to daughter. Assuming one genome per nucleoid, what can be inferred about the contribution of shuffling due to fission fusion cycles based on the author's modelling?

According to our simulations, the $ndau=11$ and $nspl=5$ combination would best match the experimentally determined rate of variant segregation. We now mention this number in the text. However, as stated in the discussion, the exact amount of fission events predicted by our model should be treated carefully, as our simplistic model does not take into account parameters such as the branched nature of the mitochondrial network, nor a link between mtDNA replication and fission, which likely influence the speed of segregation.

Furthermore, we have now included experiments (Fig. 5G and S13) to assess the contribution of fission and fusion on the rate of mtDNA variant segregation. As predicted by our model, a lack of mitochondrial fission in $\Delta dnm1$ cells results in faster variant segregation, most likely due to decreased shuffling.

Reviewer #2:

... While these are important findings, the authors could have used this approach to explore further aspects of mtDNA inheritance, such as tracking the dynamics of mutated versus wildtype mtDNA. This would in my view significantly increase novelty and the impact of the study.

To increase the novelty of our manuscript we have applied our experimental setup to examine the role of mitochondrial fission and the impact of increased mtDNA copy number on mtDNA variant

segregation. Our experiments demonstrate that absence of mitochondrial fission accelerates, whereas increased mtDNA copy number delays mtDNA variant segregation (Fig. 5G, S10). Additionally, we now demonstrate that our approach also allows assessment of mtDNA variant segregation in a setting where intact mtDNA competes against mutant mtDNA. In these experiments, we observe purifying selection against mutant mtDNA lacking the *COB* gene. We believe that these experiments greatly enhance the impact of our study.

1. From the description in the main text and methods section it is not clear if/how the authors have validated the automated tracking of cell over multiple generations. Where at least some of the data also manually annotated and resulted in the same lineage tree?

The automated tracking was always manually checked and corrected, if necessary. We have added a clarifying sentence in the method section.

2. What is visualized is actually not the mtDNA itself, but rather the expressed protein (and its redistribution to the progenies). Although the authors perform experiments to show that the decay time for both Atp6-NG and Atp6-mKate2 are similar, the half-life time seems to be in the range of hours. Therefore the loss of heteroplasmy might be significantly underestimated. Could this be corrected for in their model?

We thank the reviewer for this suggestion. Currently, the model is based on 0s and 1s that represent mtDNA copies. As pointed out by the reviewer, in the experiments fluorescently tagged Atp6 is used as a proxy for mtDNA itself. To account for this in the model, parameters such as mtDNA expression and protein stability would need to be introduced. Furthermore, it has been proposed that mitochondrial biogenesis occurs to different extents in the mother and the daughter cell (Rafelski et al., Science, 2012), and it is therefore unclear if existing proteins are distributed equally between mother and daughter. It will be interesting to further develop the mathematical model. However, we feel that the introduction of more parameters would complicate the model at this point. We, however, emphasize the point that we likely underestimate the rate of variant segregation in the revised version of the manuscript.

3. It is not clear to me why the wildtype mtDNA-Atp6-NG strain increases fluorescence over time, which the authors attribute to the switch in media, while the wildtype mtDNA-Atp6mKate2 does not show this increase upon media switch (and even display a decrease in signal)?

We agree that this finding is unexpected. Unfortunately, we currently cannot pinpoint the cause for this effect. We assume that this effect is caused by the switch from rich to minimal medium, which may affect mtDNA expression and the biophysical properties of the fluorescent proteins, as well as differences in the maturation time between NG and mKate2. However, this difference does not affect the quantification of heteroplasmy in our experiments, since NG and mKate2 fluorescence is normalized to the median fluorescence intensity per timeframe of the respective channel, in all our segregation experiments. We emphasize this point in the revised manuscript. We feel that determining the precise cause for the difference in Atp6-NG or Atp6-mKate2 over time would not add significant new insight.

4. When estimating the percentage of hetero- and homoplasmic populations (Fig. 3), autofluorescence in the "absent" channel is given as a potential explanation. It is surprising that such high levels of

autofluorescence appear in healthy cells. I would suggest to compare the levels measured with wildtype yeast, which do not express any fluorescent protein, to verify if equally high levels of autofluorescence are measured.

To address this reviewer's comment, we have assessed 'green' or 'red' fluorescence signal, representing background and/or bleed-through signal, in haploid cells exclusively expressing Atp6-mKate2 or Atp6-NG, respectively. We were able to perform these measurements on the timelapse images that we used to record segregation of mtDNA variants in heteroplasmic populations, since non-mated haploid cells were occasionally present in the field-of-view. The h-values determined for such cells is comparable to diploid cells, which we had classified as homoplasmic. Data representing these results are now shown in Fig. S7.

5. Regarding the effect of fission and fusion on mtDNA inheritance, it has been shown previously, that mitochondrial fission is also linked with mtDNA replication. Is this accounted for in the model (and if not, why do the authors think it is neglectable)?

We thank the reviewer for raising this point. Unfortunately, incorporating into the model that there is a bias for a mitochondrial fission event in-between replicated mtDNA copies would have required extensive re-writing of the code underlying our simulations, which was not feasible given our available resources and time constraints. We now acknowledge in the manuscript that we do not consider a link between mtDNA replication and fission instances. We aim to further refine the model in future studies to take into account further parameters, such as the relationship between mtDNA replication and fission and the branched nature of the mitochondrial network.

Reviewer #3:

1. Is there mixing of mtDNA variants? The transition from heteroplasmy to homoplasmy in the population is rapid. Given this timeframe, it is possible that the fluorescence is free diffusing across the network, but the mtDNA (nucleoid) variants in the zygote do not mix. Authors should assess whether there is mixing, or if the mtDNA (nucleoids) remain spatially segregated.

We thank the reviewer for this comment. We have now provided more information on this matter in the revised version of the manuscript. On one hand we refer to previous work which has established that medial buds receive mtDNA from both parental strains (Strausberg and Perlman, Mol Gen Genet., 1978), while lateral buds inherit predominantly only one mtDNA variant from the parental cell from which the bud emerges. On the other hand, we also provide an analysis, where we determine h-values in medial and lateral buds (Fig. S6A) and an analysis of segregation kinetics, where all data derived from lateral buds are excluded (Fig. S6B). Importantly, excluding the lateral buds does not dramatically change the segregation kinetics compared to the dataset including the lateral buds.

2. If there is no DNA mixing, is the present experimental setup truly reflective of heteroplasmy (as one would expect in physiological conditions)? A scenario where there is no mtDNA mixing could also result population transition towards homoplasmy, as at each division, cells would predominantly inherit only one of the two DNA variants.

Please refer to response to previous comment.

3. Following on the above comment, does the model assume that there is mixing between the mtDNA variants in the zygote? How would the segregation dynamics play out in a scenario where this did not occur?

In our initial manuscript, we had included simulations, where segregation starts from arrays where mtDNA variants are semi-mixed (000111000111...) or alternating (0101010...). To address the reviewer's comment, in the revised version of the manuscript we included an additional simulation that starts from a non-mixed condition (00000.....11111). As apparent in Fig. S9, the predicted segregation pattern does not drastically change.

4. The section on asymmetric partitioning needs further explanation/ analysis: For the asymmetric partitioning model to apply, daughter cells would have to inherit a fixed amount of mitochondria and mtDNA from the mother cell, early during bud formation. After this, as the daughter cell grows, mitochondria and mtDNA would have to be synthesized in the daughter cell. It is not so clear whether this is indeed the case, or that transmission of mitochondria and mtDNA occurs through the growth period of the bud up until division. Could the authors clarify this, and provide support for the scenario they are favouring? Authors should provide quantification of the number of nucleoids and cell area for mother and daughter cells at the time of division (when daughter and mother cell sizes should be comparable), and the number of nucleoids across cell area for mother and daughter cells from birth till division.

As requested by the reviewer, we have quantified mtDNA spots using the mtLacO-LacI system over the duration of a whole cell cycle with a live-cell imaging experiment. Indeed we see that the foci number drops during the first 45 minutes in the mother cell, during which we observe mitochondrial content exchange between mother and daughter cell. After 45 minutes, we find that the mtDNA foci number increases in daughter cells, indicating that mtDNA is being newly synthesized in the growing daughter cell. In contrast, no further reduction of mtDNA copy number is seen in the mother cell after 45 minutes. The figure representing these data is now presented as Fig. S12.

Best regards,

Christof and Ria

Dear Dr. Osman,

Thank you again for submitting your revised manuscript for our consideration. Based on the positive re-reviews (below) of the three original referees, we have now accepted it for publication in The EMBO Journal.

Yours sincerely,

Referee #1:

The authors have addressed all my comments.

Referee #2:

The authors have address all points raised or provided a sufficient explanation on why certain factors could not be included into their model.

Referee #3:

Authors have address my queries satisfactorily. I congratulate them on this excellent work.
